# Glutathione de novo synthesis but not recycling process coordinates with glutamine catabolism to control redox homeostasis and directs murine T cell differentiation

Gaojian Lian[1,2†], JN Rashida Gnanaprakasam[1†], Tingting Wang[1], Ruohan Wu[1], Xuyong Chen[1], Lingling Liu[1], Yuqing Shen[1], Mao Yang[3], Jun Yang[4], Ying Chen[5], Vasilis Vasiliou[5], Teresa A Cassel[6,7,8], Douglas R Green[3], Yusen Liu[9], Teresa WM Fan[6,7,8], Ruoning Wang[1]*

[1]Center for Childhood Cancer and Blood Diseases, Hematology, Oncology and BM, The Research Institute at Nationwide Children's Hospital, Ohio State University, Columbus, United States; [2]Medical Research Center, University of South China, Hengyang, Hunan Province, China; [3]Department of Immunology, St. Jude Children's Research Hospital, Memphis, United States; [4]Department of Surgery, St. Jude Children's Research Hospital, Memphis, United States; [5]Department of Environmental Health Sciences, Yale School of Public Health, Yale University, New Haven, United States; [6]Department of Toxicology and Cancer Biology, University of Kentucky, Lexington, United States; [7]Markey Cancer Center, University of Kentucky, Lexington, United States; [8]Center for Environmental and Systems Biochemistry, University of Kentucky, Lexington, United States; [9]Center for Perinatal Research, The Research Institute at Nationwide Children's Hospital, Ohio State University, Columbus, Ohio, United States

*For correspondence:
ruoning.wang@
nationwidechildrens.org

[†]These authors contributed equally to this work

Competing interests: The authors declare that no competing interests exist.

**Abstract** Upon antigen stimulation, T lymphocytes undergo dramatic changes in metabolism to fulfill the bioenergetic, biosynthetic and redox demands of proliferation and differentiation. Glutathione (GSH) plays an essential role in controlling redox balance and cell fate. While GSH can be recycled from Glutathione disulfide (GSSG), the inhibition of this recycling pathway does not impact GSH content and murine T cell fate. By contrast, the inhibition of the de novo synthesis of GSH, by deleting either the catalytic (Gclc) or the modifier (Gclm) subunit of glutamate–cysteine ligase (Gcl), dampens intracellular GSH, increases ROS, and impact T cell differentiation. Moreover, the inhibition of GSH de novo synthesis dampened the pathological progression of experimental autoimmune encephalomyelitis (EAE). We further reveal that glutamine provides essential precursors for GSH biosynthesis. Our findings suggest that glutamine catabolism fuels de novo synthesis of GSH and directs the lineage choice in T cells.
DOI: https://doi.org/10.7554/eLife.36158.001

## Introduction

Glutathione (GSH) is the most abundant antioxidant capable of providing reducing equivalents and it also serves as a versatile nucleophilic cofactor in a wide spectrum of metabolic reactions in aerobic organisms (*Kosower and Kosower, 1978*; *Meister, 1982*). While some cells are capable of

employing extracellular GSH, the utilization of extracellular GSH plays a minor role in regulating GSH homeostasis, since the extracellular levels of GSH are normally three orders of magnitude lower than intracellular GSH concentrations, which are usually in the millimolar range (*Kosower and Kosower, 1978*; *Meister, 1982*; *Perrone et al., 2005*; *Hwang et al., 1992*; *Ganguly et al., 2003*; *Bennett et al., 2009*; *Morgan et al., 2011*; *Park et al., 2016*). Hence, cellular GSH content is largely determined by intracellular production through de novo synthesis, a process mediated by two ATP-dependent ligases, glutamate-cysteine ligase (GCL) and glutathione synthase (GS), as well as through regeneration of GSH from GSSG, a process catalyzed by glutathione disulfide reductase (GSR) (*Meister, 1982*; *Lu, 2009*). In the process of de novo synthesis, GCL, a heterodimer of a catalytic subunit (GCLC) and a modifier subunit (GCLM), catalyzes the first and rate-limiting step to form the dipeptide γ-glutamylcysteine (γ-GC) from cysteine and glutamate (*Franklin et al., 2009*; *Chen et al., 2005*). After the initial step, GS catalyzes the formation of GSH by ligating γ-GC with glycine. As such, GSH synthesis is determined by the availability of its constituent amino acids, cysteine, glycine and glutamate, which intersects with glucose and glutamine metabolic pathways and reflects the overall metabolic status in the cell. In particular, glutamine catabolism may coordinate with de novo GSH synthesis by promoting cysteine uptake and providing glutamate, an immediate product of glutamine after deamination (*Altman et al., 2016*; *Hensley et al., 2013*; *Gorrini et al., 2013*). However, it remains unknown the extent to which de novo synthesis versus recycling of GSSG contributes to GSH homeostasis in T cells and how the perturbation of GSH homeostasis impacts T cell differentiation.

The regulation of metabolic pathways is tightly linked with T cell activation, differentiation, and immune functions (*Wang and Green, 2012*; *Pearce and Pearce, 2013*; *O'Neill et al., 2016*; *Patel and Powell, 2017*; *Ma et al., 2017*; *Buck et al., 2017*; *Zeng and Chi, 2017*; *Finlay and Cantrell, 2011*; *Weinberg et al., 2015*). We and others have shown that activation of T cells leads to a significant enhancement of aerobic glycolysis but a suppression of mitochondria-dependent fatty acid oxidation (FAO) (*Wang et al., 2011*; *Frauwirth et al., 2002*; *Jacobs et al., 2008*; *Shi et al., 2011*). Following the initial growth stage of T cell activation, FAO fuels and drives induced CD4[+] regulatory T (Treg) cell differentiation (*Shi et al., 2011*; *Michalek et al., 2011*). In contrast, a persistent glycolytic program is engaged not only during the initial growth phase of T cell activation but also throughout the differentiation of other CD4[+]T helper (T$_H$) cells and CD8[+] cytotoxic T (CTL) cells (*Shi et al., 2011*; *Michalek et al., 2011*; *Finlay et al., 2012*). However, oxygen consumption is also dramatically elevated in active T cells since heightened glutamine catabolism via mitochondria-dependent oxidation following T cell activation fuels oxidative phosphorylation (OXPHOS) by providing α-ketoglutarate (α-KG), an anaplerotic substrate of the tricarboxylic acid cycle (TCA cycle) (*Wang et al., 2011*; *Sena et al., 2013*; *Kamiński et al., 2012*). OXPHOS in mitochondria generates ATP through the electron transport chain (ETC) and also produces reactive oxygen species (ROS) as byproducts, rendering mitochondria a major source of intracellular ROS. Superoxide anion ($O_2^{\bullet-}$), the 'primary' ROS derived from mitochondrial ETC, is converted to hydrogen peroxide ($H_2O_2$) by spontaneous and enzymatic processes, whereupon $H_2O_2$ freely diffuses into cytosol and functions as a redox signaling molecule to elicit a diverse array of cellular responses, the spectrum of which depends on the level of ROS (*Sena et al., 2013*; *Kamiński et al., 2012*; *Murphy, 2009*; *Schieber and Chandel, 2014*). A fine-tuned balance between ROS generation and antioxidant capacity ensures physiological levels of intracellular ROS, which are required for driving essential signaling events to support T cell-mediated immune responses (*Weinberg et al., 2015*; *Murphy and Siegel, 2013*; *Kamiński et al., 2013*; *Simeoni et al., 2016*; *Rashida Gnanaprakasam et al., 2018*). Accordingly, oxidative stress, occurring when ROS generation exceeds the capacity of antioxidants, dampens essential cellular processes and functions of T cells. In innate immune cells, ROS are effector molecules that are capable of directly killing pathogens as well as act as redox signaling molecules that modulate a wide range of innate immune responses (*Schieber and Chandel, 2014*; *Mills et al., 2017*; *Nathan and Cunningham-Bussel, 2013*). Accumulating evidence has shown that ROS production is induced following T cell activation and is required for driving T cell activation and proliferation (*Sena et al., 2013*; *Kamiński et al., 2012*). Also, T$_H$17 cells are more sensitive to the damaging effects of ROS than are T$_{reg}$ cells (*Gerriets et al., 2015*). A recent study demonstrated a critical role for GSH biosynthesis in fine-tuning this process by maintaining ROS hemostasis and regulating Myc-dependent T cell metabolic reprogramming during T cell activation (*Mak et al., 2017*;

*Klein Geltink et al., 2017*). Little information exists, however, on whether and how T cell metabolic programs modulate T cell GSH biosynthesis and ROS homeostasis.

Here, we report a critical role for de novo GSH synthesis but not recycling of GSSG in modulating ROS homeostasis and T cell differentiation. Heightened glutamine catabolism during $T_H17$ differentiation provides glutamate to support de novo GSH synthesis and suppresses oxidative stress. Genetic ablation of de novo synthesis of GSH but not regeneration of GSH from GSSG leads to the augmentation of ROS, dampening $T_H17$ differentiation while enhancing $T_{reg}$ cell differentiation. Moreover, we found that dimethyl fumarate, an FDA approved drug (BG-12/Tecfidera) for multiple sclerosis, suppresses $T_H17$ differentiation by augmenting intracellular ROS. Combining pharmacological and genetic approaches, our studies implicate the GSH-ROS axis as a metabolic checkpoint coordinating glutamine catabolism and T cell signaling to direct T cell differentiation.

## Results

### De novo synthesis but not recycling of GSSG is required for producing GSH and suppressing ROS upon TCR stimulation

T cell activation is associated with enhanced GSH and ROS production (*Figure 1—figure supplement 1A and B*) (*Sena et al., 2013*; *Kamiński et al., 2012*; *Mak et al., 2017*). GSH can be regenerated through recycling of glutathione disulfide (GSSG) or synthesized de novo from glutamate, cysteine and glycine (*Figure 1A*). GSH regeneration is mediated by glutathione-disulfide reductase (GSR), whereas de novo synthesis is composed of two steps catalyzed by glutamine-cysteine ligase (GCL), a heterodimer of a catalytic subunit (GCLC) and a modulatory subunit (GCLM), and glutathione synthase (GS), respectively (*Figure 1A*). Thus, we examined the expression of the above key enzymes following T cell activation. Real time quantitative PCR (qPCR) analysis revealed a time-dependent up-regulation of mRNAs encoding these metabolic enzymes in T cells following activation (*Figure 1—figure supplement 1C*).

To determine the extent to which de novo synthesis contributes to GSH production and redox homeostasis in T cells, we obtained mouse models with genetic deficiencies in GCL. GCLC possesses all the enzymatic activity, while GCLM functions to optimize the catalytic efficiency of the holoenzyme (*Chen et al., 2005*). *Gclm* knockout (*Gclm* KO) mice carry the germ-line deletion of *Gclm*, whereas T cell-specific *Gclc* knockout (T cell-*Gclc* KO) mice, generated by crossing *Gclc-floxed* mice with CD4-Cre mice, carry the *Gclc* deletion exclusively in T cells (*Chen et al., 2007*; *Yang et al., 2002*). Absent expression of GCLM or GCLC in T cells derived from corresponding animals was confirmed by western blot (*Figure 1—figure supplement 1D*). Next, we examined the intracellular levels of GSH and ROS of T cells that were stimulated with anti-CD3 plus anti-CD28. Deficiency in GCLC (the catalytic subunit) and, to a lesser extent, deficiency in GCLM (modifier subunit) resulted in reduced intracellular content of GSH (*Figure 1B*). Consistent with this, we observed increased ROS in *Gclc-* and to a lesser extent in *Gclm*-deficient T cells as compared to WT T cells. (*Figure 1C*).

We then sought to determine the extent to which recycling of GSSG contributes to GSH production and redox homeostasis in T cells. For this, we obtained mice carrying germ-line deletion of *Gsr* (*Gsr−/−*), the deletion of which was demonstrated by qPCR (*Figure 1—figure supplement 1D*) (*Rogers et al., 2004*; *Pretsch, 1999*; *Yan et al., 2012*). However, WT and *Gsr*-deficient T cells displayed comparable GSH and ROS levels (*Figure 1B and C*). These results suggested that de novo synthesis of GSH by the metabolic pathway plays an indispensable role in producing GSH and maintaining redox homeostasis during T cell activation.

### GCLC deficiency but not GCLM or GSR deficiency suppresses T cell activation and proliferation

*Gclm* KO, T cell-*Gclc* KO and *Gsr* KO mice contained comparable numbers and distribution of thymocytes and peripheral CD4+ and CD8+ T cells relative to control mice (*Figure 2—figure supplement 1A,B,C and D*), indicating a largely undisturbed T cell development and distribution after double positive stage in the absence of GSH recycling pathway or the de novo synthesis pathway. A recent study has showed that Gclc deficiency suppressed T cell activation and proliferation, demonstrating a critical role of GCLC in regulating T cell activation (*Mak et al., 2017*; *Klein Geltink et al., 2017*). This is consistent with the severe GSH depletion and ROS induction in *Gclc*-deficient active T

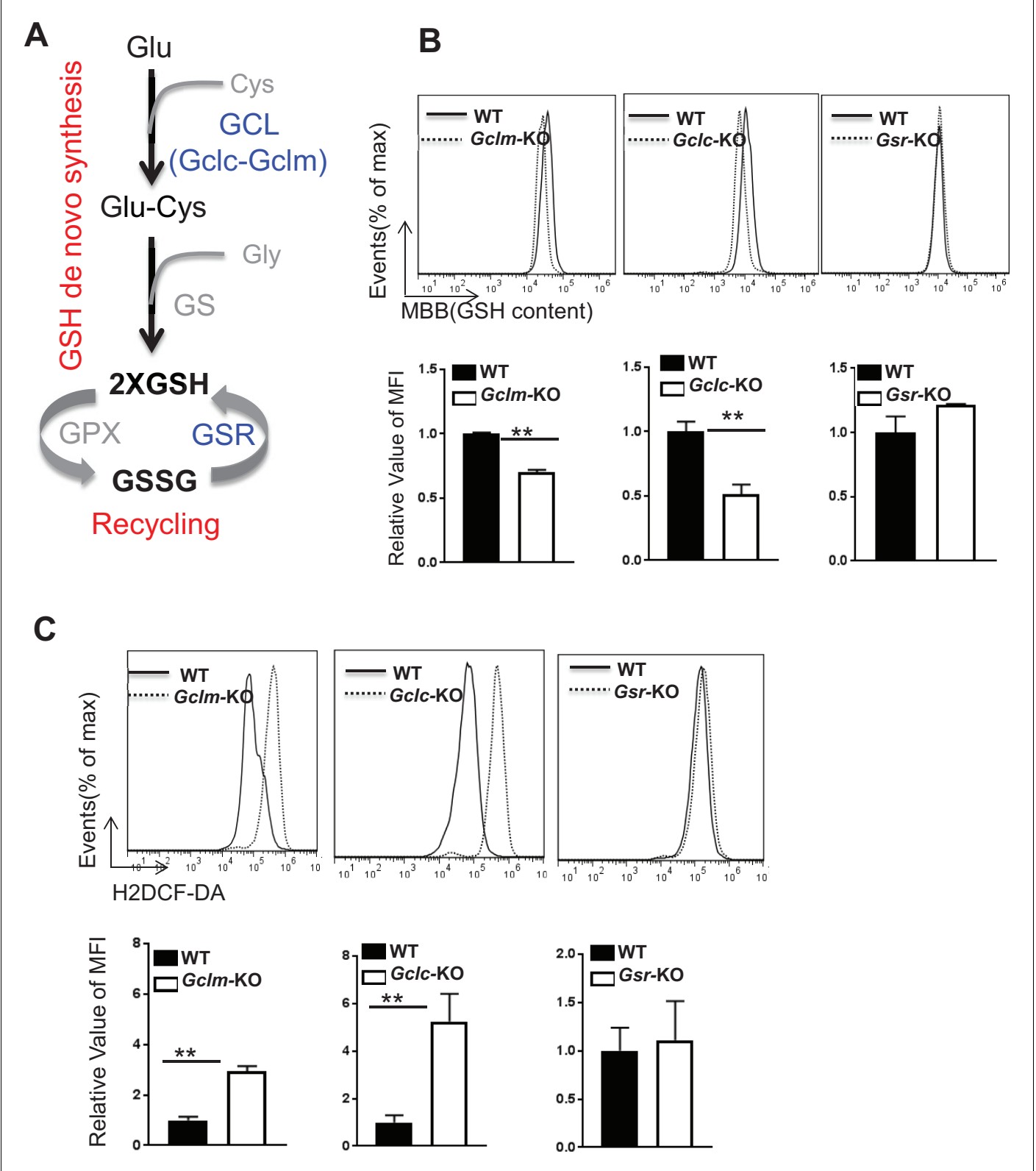

**Figure 1.** De novo synthesis but not recycling of GSSG is required for producing GSH and fine-tuning ROS upon TCR stimulation. (A) Diagram of GSH biosynthesis, with metabolic pathways highlighted in red and enzymes highlighted in blue. (B) Naive CD4+T cells from WT and *Gclm* KO (left), or WT (*CD4-Cre-, Gclc*fl/fl) and *Gclc* KO (*CD4-Cre+, Gclc*fl/fl, (middle), or WT and *Gsr* KO (right) were activated by plate-bound anti-CD3 plus anti-CD28 for 24 hr, followed by the measurement of GSH levels. (C) Naive CD4+T cells from WT and *Gclm* KO (left), or WT (*CD4-Cre-, Gclc*fl/fl) and *Gclc* KO (*CD4-Cre+,*

*Figure 1 continued on next page*

*Figure 1 continued*

*Gclc*^fl/fl^, (middle), or WT and *Gsr* KO (right) were activated by plate-bound anti-CD3 plus anti-CD28 for 24 hr, followed by the measurement of ROS levels. Data in *Figure 1B–C* are representative of two independent experiments. Data represent the mean ± S.D.

DOI: https://doi.org/10.7554/eLife.36158.002

The following source data and figure supplements are available for figure 1:

**Source data 1.** Source data for B and C.

DOI: https://doi.org/10.7554/eLife.36158.005

**Figure supplement 1.** TCR stimulation drives GSH and ROS production in T cells.

DOI: https://doi.org/10.7554/eLife.36158.003

**Figure supplement 1—source data 1.** Source data for A, B, C and D.

DOI: https://doi.org/10.7554/eLife.36158.004

cells (*Figure 1B and C*). Our results further suggested that GSR is dispensable and GCLM only play a minor role in modulating GSH production and ROS homeostasis in active T cells (*Figure 1B and C*). Next, we sought to differentiate the impact of GSH recycling pathway and de novo synthesis pathway on T cell activation and proliferation. While both Gclc and Gclm deficiency caused a reduction of GSH and induction of ROS, albeit to different degrees (*Figure 1B and C*), Gclc but not Gclm deficiency resulted in an impairment of cell viability, appearance of the activation marker CD25, and activation-induced cell proliferation (*Figure 2A, B, D, E and G*). This activation and proliferation defect is consistent with a recent study showing that GCLC is required for T cell activation (*Mak et al., 2017*). Previous studies have shown that GSH depletion caused inactivation of glutathione peroxidase 4 (GPX4), and consequentially led to iron-dependent accumulation of lipid peroxidation and a form of cell necrosis referred as ferroptosis (*Stockwell et al., 2017*). Our data suggested that a moderate reduced cell viability in *Gclc*-deficient T cells is likely due to ferroptosis in the context of GSH depletion (*Figure 2G*). Our results suggested that a severe depletion of GSH and induction of ROS caused by Gclc deficiency significantly impaired T cell activation and proliferation, however, active T cell could tolerate and cope with a moderate depletion of GSH and induction of ROS caused by Gclm deficiency.

Consistent with its dispensable role in GSH production in active T cells (*Figure 1B and C*), Gsr deficiency did not lead to any impairment of T cell viability, activation marker CD25 and proliferation (*Figures 2C, F and G*). Notably, none of the above genetic deficiencies compromised the early activation marker CD69 (*Figure 2A–C*). Our results suggest that GSH recycling pathway is dispensable in regulating T cell activation and proliferation. However, the recycling of GSSG to GSH, which is not a parallel pathway for GSH production, plays a critical role in maintaining redox homeostasis when the ratio of GSSG:GSH reaches certain threshold. While our data suggest that Gsr mediated recycling of GSSG is not required for T cell activation and proliferation, we do not have evidence showing that GSSG is significantly accumulated during T cell activation and proliferation. Therefore, a dispensable role of Gsr in maintaining redox homeostasis during T cell activation and proliferation may represent a context-dependent interpretation.

## Ablation of de novo synthesis but not recycling of GSSG reciprocally alters T$_H$17 and iT$_{reg}$ cell differentiation

Following the initial growth stage of cell activation, proliferating CD4$^+$ T cells can differentiate into various functional subsets including inflammatory T$_H$17 and Foxp3-expressing regulatory T cells (T$_{reg}$ cells), which are two closely related subsets with distinct functions. Having found a role for de novo synthesis of GSH in modulating GSH and ROS homeostasis during T cell activation, we next assessed the functional requirement for the de novo synthesis pathway of GSH in T cell differentiation. Naive T cells were differentiated under T$_H$17 or iT$_{reg}$ conditions. As compared to control WT cells, *Gclm* KO cells exhibited reduced IL-17$^+$ and increased Foxp3$^+$ cells (*Figure 3A and D*). Given that a similar degree of proliferation was observed between WT and *Gclm* KO CD4$^+$ T cells (*Figure 3A and D*), deregulation of cell differentiation in *Gclm* KO was largely independent of cell expansion. To bypass the effect of Gclc deficiency on T cell activation and proliferation (*Figure 2B and E*), we generated a mouse model carrying a conditional *Gclc* allele (*Gclc*^flox/flox^) and a tamoxifen-induced Cre recombinase (CreERT2) transgene (*Ryan et al., 2000*; *de Luca et al., 2005*), which allowed us to delete *Gclc*

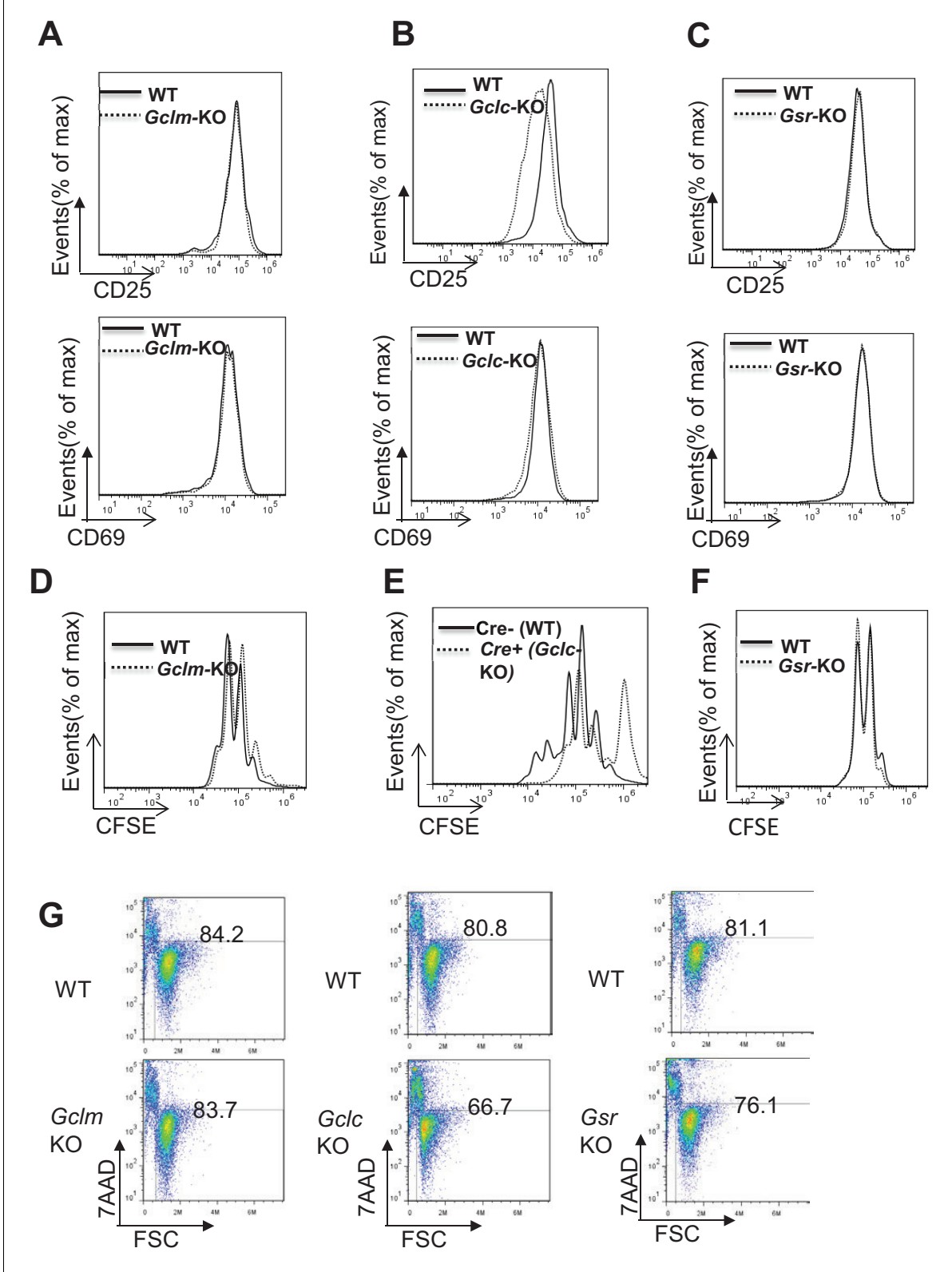

**Figure 2.** Severe depletion of GSH by blocking de novo synthesis suppresses T cell activation and proliferation. (A–C) Naive CD4 +T cells from WT and *Gclm* KO (A), or WT (*CD4-Cre-, Gclc^{fl/fl}*) and *Gclc* KO (*CD4-Cre+, Gclc^{fl/fl}*) (B), or WT and *Gsr* KO (C) mice were activated by plate-bound anti-CD3 plus anti-CD28 for 24 hr, followed by cell surface expression of CD25 (upper panel) and CD69 (lower panel). (D–F) Cell proliferation of active CD4 +T cells (72 hr) with indicated genotypes was determined by CFSE dilution. (G) Naive CD4 +T cells isolated from mice with indicated genotypes were activated

*Figure 2 continued on next page*

*Figure 2 continued*

by plate-bound anti-CD3 and anti-CD28 for 24 hr. Cell viability was determined by FACS. *Figure 2A–G* are representative of three independent experiments.

DOI: https://doi.org/10.7554/eLife.36158.006

The following source data and figure supplements are available for figure 2:

**Figure supplement 1.** Severe depletion of GSH by blocking de novo synthesis suppresses T cell activation and proliferation.

DOI: https://doi.org/10.7554/eLife.36158.007

**Figure supplement 1—source data 1.** Source data for A, B, C and D.

DOI: https://doi.org/10.7554/eLife.36158.008

flox alleles in an acute manner. For this, we polarized T cells in the absence (WT) or in the presence (KO) of 4-Hydroxytamoxifen (4OHT) (*Figure 3—figure supplement 1A*). As compared with WT cells, acute deletion of Gclc bypassed its required for cell proliferation, as revealed by a comparable CFSE dilution (*Figure 3B and E*), but nevertheless resulted in a reduction in the generation of IL-17$^+$ T cells and an induction in the generation of Foxp3$^+$ cells. Using this genetic model, we have therefore differentiated the role of GCLC in early T cell activation from its role in driving T$_H$17 cell differentiation. In contrast to the effects of ablation of *Gclm* and *Gclc*, *Gsr* KO and WT CD4$^+$ T cells displayed a similar degree of T$_H$17 and iT$_{reg}$ differentiation, indicating a dispensable role for recycling GSSG in regulating T cell differentiation (*Figure 3C and F*). Next, we tested whether inhibiting ROS generation in T cells that have defects on the de novo synthesis of GSH would restore T$_H$17 differentiation. The addition of N-acetyl-L-cysteine (NAC), a reagent often used to scavenge ROS, restored T$_H$17 cell differentiation in *Gclc$^{-/-}$* T cells (*Figure 3—figure supplement 1B*). While NAC is frequently considered a source of cysteine for synthesis of GSH, while other studies have shown that NAC displayed reducing properties through its thiol-disulfide exchange activity and could directly scavenge free radicals (*Ates et al., 2008*; *Agnihotri and Mishra, 2009*; *Zafarullah et al., 2003*; *Cotgreave, 1997*). Consistent with these, our result suggests that NAC scavenges ROS independently of GSH synthesis (*Figure 3—figure supplement 1C and D*) and therefore restores T$_H$17 cell differentiation in *Gclc$^{-/-}$* T cells. To further evaluate the role of the GSH de novo synthesis pathway in T$_H$17-driven EAE in vivo, we immunized mice with the myelin oligodendrocyte glycoprotein (MOG)$_{35-55}$ antigen. Genetic ablation of GCLM or GCLC conferred a protection against disease progression (*Figure 3G and H*). Furthermore, histological assessment revealed a similar degree of ablation of T cell infiltration in *Gclm* KO and T cell-*Gclc* KO animals compared to WT animals (*Figure 3—figure supplement 1E*). However, a different degree of ablation of macrophage infiltration in *Gclm* KO and T cell-*Gclc* KO animals compared to WT animals was observed and likely reflected a different level of inflammation in these experimental animals (*Figure 3—figure supplement 1E*). In contrast to T cell-*Gclc* KO animals, GCLM is also deleted in macrophages in *Gclm* KO animal (a germline KO model). Therefore, we could not exclude the possibility that Gclm deficiency in macrophages might affect macrophage infiltration in our result. Collectively, our studies suggested that de novo synthesis of GSH is required for T$_H$17 development in vitro and T$_H$17-driven CNS inflammation in vivo.

## T$_H$17 and iT$_{reg}$ cells display different degrees of oxidative stress

Accumulating evidence has shown that each subset of T cells engages unique metabolic pathways to fulfill its metabolic demands (*Shi et al., 2011*; *Michalek et al., 2011*; *Gerriets et al., 2015*). We envisioned that the differential engagement of metabolic pathways would differentially impact GSH biosynthesis and cellular oxidative stress in T$_H$17 and iT$_{reg}$ cells. For this, we activated naive CD4$^+$ T cells under T$_H$17 or iT$_{reg}$ polarizing conditions in vitro, and examined the intracellular levels of GSH and ROS at day 3 and day 5. T$_H$17 cells displayed a higher level of intracellular GSH than iT$_{reg}$ cells. In contrast, the level of ROS was lower in T$_H$17 cells compared to iT$_{reg}$ cells (*Figure 4A and B*). In addition, T$_H$17 cells displayed a higher level of intracellular GSH and GSSG than iT$_{reg}$ cells, as revealed by mass spectrometry (*Figure 4C*). A key cellular mechanism in defending against oxidative stress is through activation of nuclear factor erythroid 2-related factor 2 (NRF2), which controls the expression of genes involved in producing, regenerating, and utilizing GSH. NRF2 also controls other antioxidant pathways that regulate thioredoxin (*TXN*), NADPH generation and iron sequestration (*Jaramillo and Zhang, 2013*; *Sporn and Liby, 2012*; *Ma, 2013*; *Motohashi and Yamamoto,*

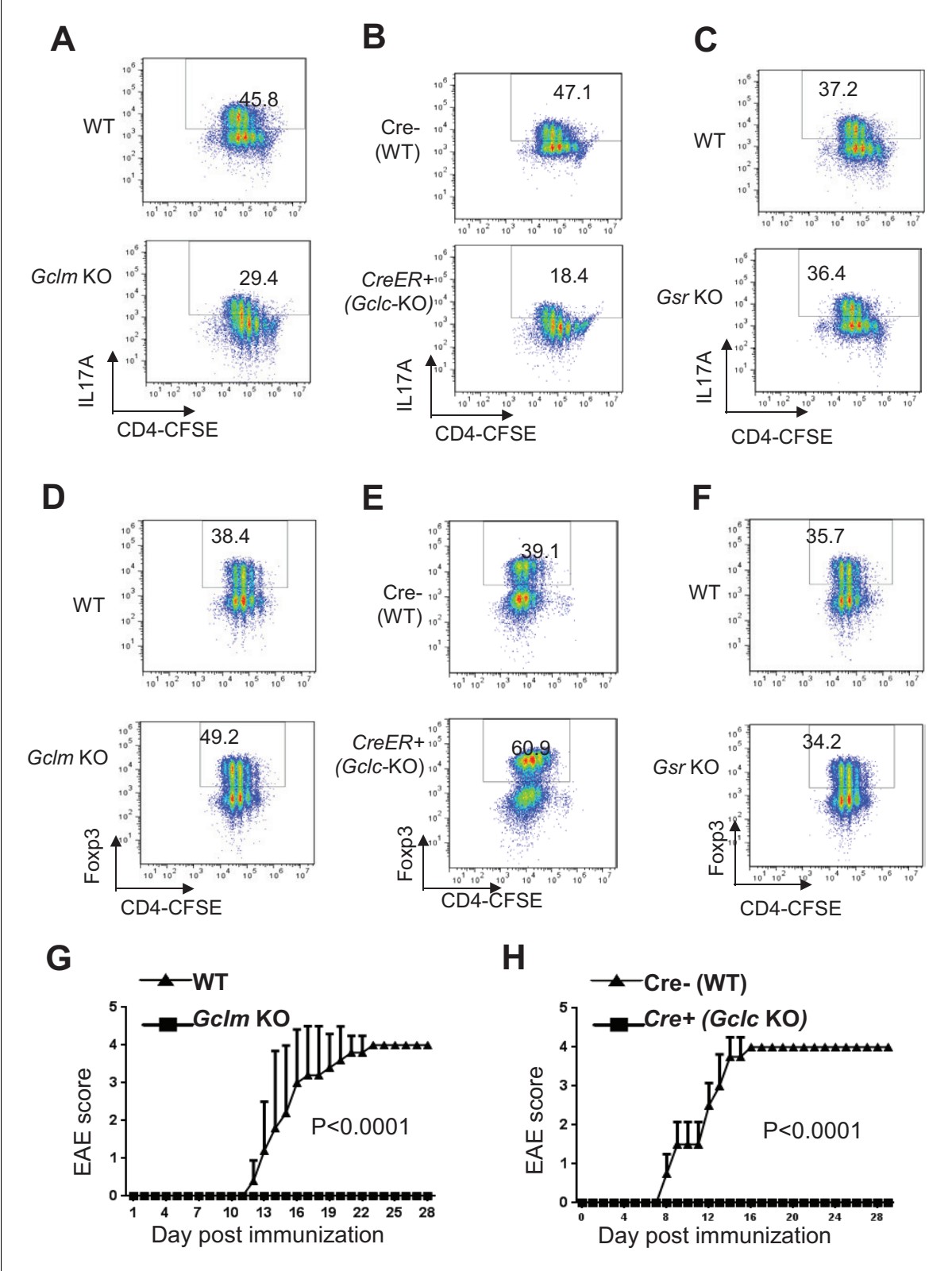

**Figure 3.** Ablation of de novo synthesis but not recycling of GSSG reciprocally alters $T_H17$ and iTreg cell differentiation. (A–F) Naive CD4+ T cells from WT and *Gclm* KO, or WT (*Cre-,*) and CreER+ (*Gclc*-KO- in the presence of 100 nM of 4-hydroxytamoxifen (4OHT)), or WT and *Gsr* KO mice were stained with 4 µm CFSE and differentiated under $T_H17$ or iTreg -inducing conditions for 5 days, followed by intracellular staining of IL-17 and Foxp3. (G–H) mice
*Figure 3 continued on next page*

*Figure 3 continued*

with indicated genotypes were immunized with MOG to induce EAE and pathological progressions were scored daily. Data in *Figure 4A–H* are representative of two-three independent experiments.

DOI: https://doi.org/10.7554/eLife.36158.009

The following source data and figure supplements are available for figure 3:

**Source data 1.** Source data for G and H.

DOI: https://doi.org/10.7554/eLife.36158.012

**Figure supplement 1.** Ablation of de novo synthesis but not recycling of GSSG reciprocally alters TH17 and iTreg cell differentiation.

DOI: https://doi.org/10.7554/eLife.36158.010

**Figure supplement 1—source data 1.** Source data for C and D.

DOI: https://doi.org/10.7554/eLife.36158.011

*2004*). Consistent with increased levels of GSH and decreased levels of ROS in $T_H17$ cells, qPCR analysis revealed a time-dependent up-regulation of mRNAs encoding NRF2 and its target genes, including glucose-6-phosphate dehydrogenase (*G6PD*), *TXN*, thioredoxin reductase 1 (*Txnrd1*), CD44, heme oxygenase-1 (*Hmox1*), NAD(P)H quinone dehydrogenase 1 (*Nqo1*) and glutathione synthetase (*Gss*) (*Figure 4—figure supplement 1A*). These results suggested that $T_H17$ cells preferentially maintain a low degree of oxidative stress by a tight regulation of GSH synthesis and ROS homeostasis.

## De novo synthesis but not recycling of GSSG is required for producing GSH and suppressing ROS during $T_H17$ cell differentiation

The observation that $T_H17$ cells displayed increased GSH content and decreased levels of ROS than $iT_{reg}$ cells supported the role for GSH in directing $T_H17$ cell differentiation (*Figure 4A and B*). Next, we sought to determine how two GSH synthetic pathways are engaged and impact on GSH and ROS homeostasis during $T_H17$ cell differentiation. For this, we purified naive T cells and differentiated them under $T_H17$-polarizing conditions. Deficiency in Gclc and, to a lesser extent, deficiency in Gclm resulted in reduced intracellular content of GSH (*Figure 4D*). Consistent with this, we observed increased ROS in *Gclc*- and *Gclm*-deficient T cells as compared to WT T cells (*Figure 4—figure supplement 1B and C*). By contrast, WT and *Gsr*-deficient T cells displayed comparable GSH sand ROS levels (*Figure 4—figure supplement 1D*). These results indicated that the preferential requirement for de novo synthesis of GSH during the initial T cell activation stage is extended to the later T cell differentiation stage. However, the recycling pathway is dispensable in producing GSH and maintaining redox homeostasis during $T_H17$ differentiation.

## Pharmacological augmentation of ROS reciprocally modulates $T_H17$ and $iT_{reg}$ cell differentiation

We next asked whether shifting the redox balance towards an oxidative state would perturb T cell differentiation and represent a novel therapeutic strategy for T cell-driven autoimmunity. For this, we activated naive $CD4^+$ T cells under $T_H17$ or $iT_{reg}$ polarizing conditions in the presence or absence of $H_2O_2$. Addition of 1 μM $H_2O_2$ did not impact cell proliferation but reciprocally reduced $T_H17$ and enhanced $iT_{reg}$ cell differentiation (*Figure 5A and B*), indicating that its effect on differentiation was largely independent of expansion.

The observation of $H_2O_2$-dependent suppression of $T_H17$ differentiation prompted us to explore pharmacologic approaches that could augment ROS production in T cells. Dimethyl fumarate (DMF), the key active ingredient of BG-12/TECFIDERA and FUMADERM, has been approved in many countries for treating autoimmune diseases including multiple sclerosis (MS) and psoriasis, both of which are associated with pathogenic $T_H17$ cells (*Study group et al., 2014*; *Mrowietz et al., 2007*; *DEFINE Study Investigators et al., 2012*; *CONFIRM Study Investigators et al., 2012*). However, the cellular and molecular mechanisms underlying the therapeutic efficacy of DMF have not been fully elucidated (*Kees, 2013*). Previous studies have implicated DMF in regulating the cellular activities of dendritic cells (DCs), endothelial cells, and neurons through various mechanisms (*Blewett et al., 2016*; *Schulze-Topphoff et al., 2016*; *Wang et al., 2015*; *Chen et al., 2014*; *Peng et al., 2012*; *Linker and Gold, 2013*; *Duffy et al., 1998*; *Gill and Kolson, 2013*). The

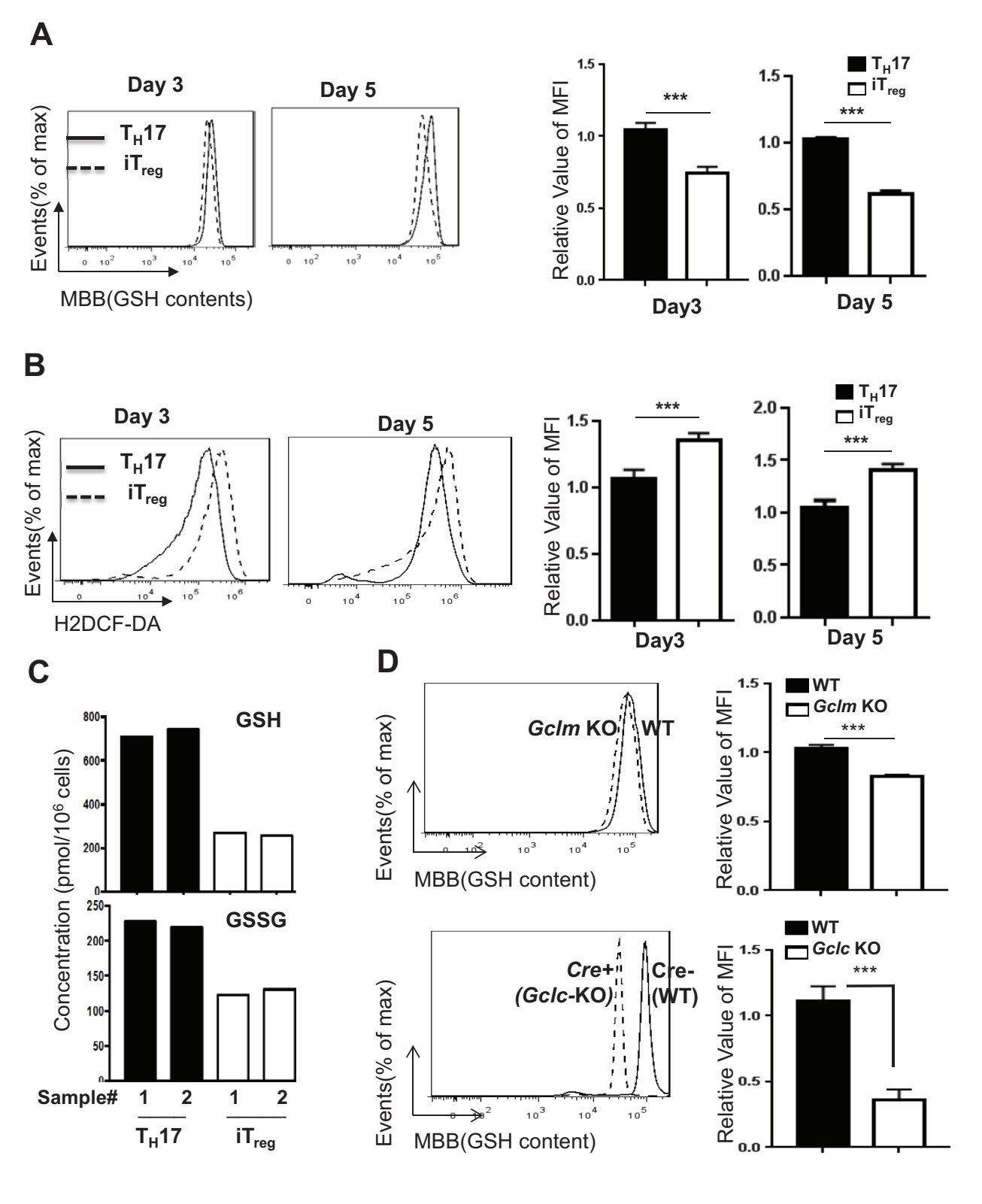

**Figure 4.** $T_H17$ cells preferentially maintain higher level of GSH than $iT_{reg}$ cells. (**A–B**) Naive CD4+ T cells from C57BL/6 mice were differentiated under iTreg or $T_H17$–inducing conditions and cells were collected at indicated times, followed by measuring intracellular GSH (**A**) and ROS (**B**) by FACS. (**C**) Naive CD4+ T cells from C57BL/6 mice were differentiated under $T_H17$ or $iT_{reg}$–inducing conditions for 5 days. The intracellular levels of GSH and GSSG were determined by mass spectrometry. (**D**) Naive CD4+T cells from WT and *Gclm* KO (top) or WT (*CD4-Cre-, Gclc^{fl/fl}*) and *Gclc* KO (*CD4-Cre+, Gclc^{fl/fl}*,

*Figure 4 continued on next page*

*Figure 4 continued*

(bottom) were differentiated under T$_H$17-inducing conditions for 5 days, followed by the measurement of GSH levels. Data in *Figure 4A–D* are representative of three independent experiments. Data represent the mean ± S.D.

DOI: https://doi.org/10.7554/eLife.36158.013

The following source data and figure supplements are available for figure 4:

**Source data 1.** Source data for A, B, C and D.

DOI: https://doi.org/10.7554/eLife.36158.016

**Figure supplement 1.** De novo synthesis but not recycling of GSSG is required for providing GSH and suppressing ROS during TH17 cell differentiation.

DOI: https://doi.org/10.7554/eLife.36158.014

**Figure supplement 1—source data 1.** Source data for A, B and C.

DOI: https://doi.org/10.7554/eLife.36158.015

electrophilic nature of DMF allows it to bind and deplete intracellular GSH (*Zheng et al., 2015*; *Sullivan et al., 2013*). We therefore hypothesized that DMF may induce oxidative stress and affect T cell differentiation. To test this, we activated naive CD4$^+$ T cells under T$_H$17 polarizing conditions in the presence of a range of DMF doses based on previous reports that were designed to investigate the effect of DMF on other cell types (*Peng et al., 2012*; *Linker and Gold, 2013*; *Duffy et al., 1998*). DMF treatment displayed a dosage-dependent suppression of T$_H$17 cell differentiation (*Figure 5C*). Although the addition of 75 µM DMF inhibited T cell proliferation, at the lower dose of DMF used in this study (20–50 µM), we observed minimal inhibitory effects of DMF on cell proliferation (*Figure 5—figure supplement 1A*), indicating that DMF-mediated suppression on T$_H$17 differentiation was largely independent of cell proliferation. Moreover, DMF treatments induced ROS production in T$_H$17 cells (*Figure 5D*). Consistent with the idea that DMF suppresses T$_H$17 cell differentiation through the induction of oxidative stress, the addition of N-acetyl-L-cysteine (NAC) restored T$_H$17 cell differentiation in the presence of DMF (*Figure 5E*). Since H$_2$O$_2$ treatment reciprocally reduced T$_H$17 and enhanced iT$_{reg}$ cell differentiation (*Figure 5A and B*), we next assessed the effect of DMF on iT$_{reg}$ cell differentiation. However, the doses of DMF (5 and 20 µM) that could suppress T$_H$17 differentiation failed to affect iT$_{reg}$ cell differentiation in vitro (*Figure 5—figure supplement 1B*). As such, our data suggested that DMF may partially exert its immunomodulatory action through the augmentation of oxidative stress and suppressing T$_H$17 differentiation. Along with the possibility that DMF-derived fumarate interferes with TCA cycle intermediate metabolite pool, other mechanisms may also contribute to its immunomodulatory functions (*Blewett et al., 2016*; *Schulze-Topphoff et al., 2016*; *Wang et al., 2015*; *Chen et al., 2014*; *Peng et al., 2012*; *Linker and Gold, 2013*; *Duffy et al., 1998*; *Gill and Kolson, 2013*).

## Glutamate that fuels GSH de novo synthesis is partially derived from glutamine during T$_H$17 cell differentiation

Upon activation, a metabolic reprogramming is required for directing nutrients to meet the bioenergetic, biosynthetic, and redox demands, which prepares T cells for immune defense and regulation. We and others have shown that T cell metabolism changes from relying on FAO and some mitochondria-dependent glucose oxidation to engaging robust aerobic glycolysis and glutaminolysis (*Wang et al., 2011*; *Frauwirth et al., 2002*; *Jacobs et al., 2008*; *Gerriets and Rathmell, 2012*; *Pearce et al., 2009*). Glutamine catabolism not only fuels mitochondrial ATP production through the TCA cycle but also provides metabolic precursors for multiple biosynthetic pathways, including synthesis of glutathione (GSH), an essential cellular antioxidant system to maintain redox homeostasis during T cell activation (*Altman et al., 2016*; *Hensley et al., 2013*; *Gorrini et al., 2013*; *Mak et al., 2017*). The de novo synthesis of GSH requires glycine, cysteine and glutamate as metabolic precursors. We envisioned that glutamine-derived glutamate partially fulfils the requirement of de novo synthesis of GSH in T$_H$17 cells (*Figure 6A*). To test this hypothesis, we followed U-$^{13}$C,$^{15}$N labeled glutamine incorporation into glutamate and GSH. The $^{13}$C$_5$ isotopologues (generated via a direct Glutamine to Glutmate conversion) represented a significant fraction of the total glutamate and GSH pool in both T$_H$17 and iT$_{reg}$ cells, however, the absolute quantity of $^{13}$C$_5$-glutamate and $^{13}$C$_5$-GSH is higher in T$_H$17 cells than in iT$_{reg}$ cells (*Figure 6B*). These results suggested that T$_H$17 cells could uptake more glutamine and/or produce more glutamate for GSH synthesis than iT$_{reg}$ cells. In

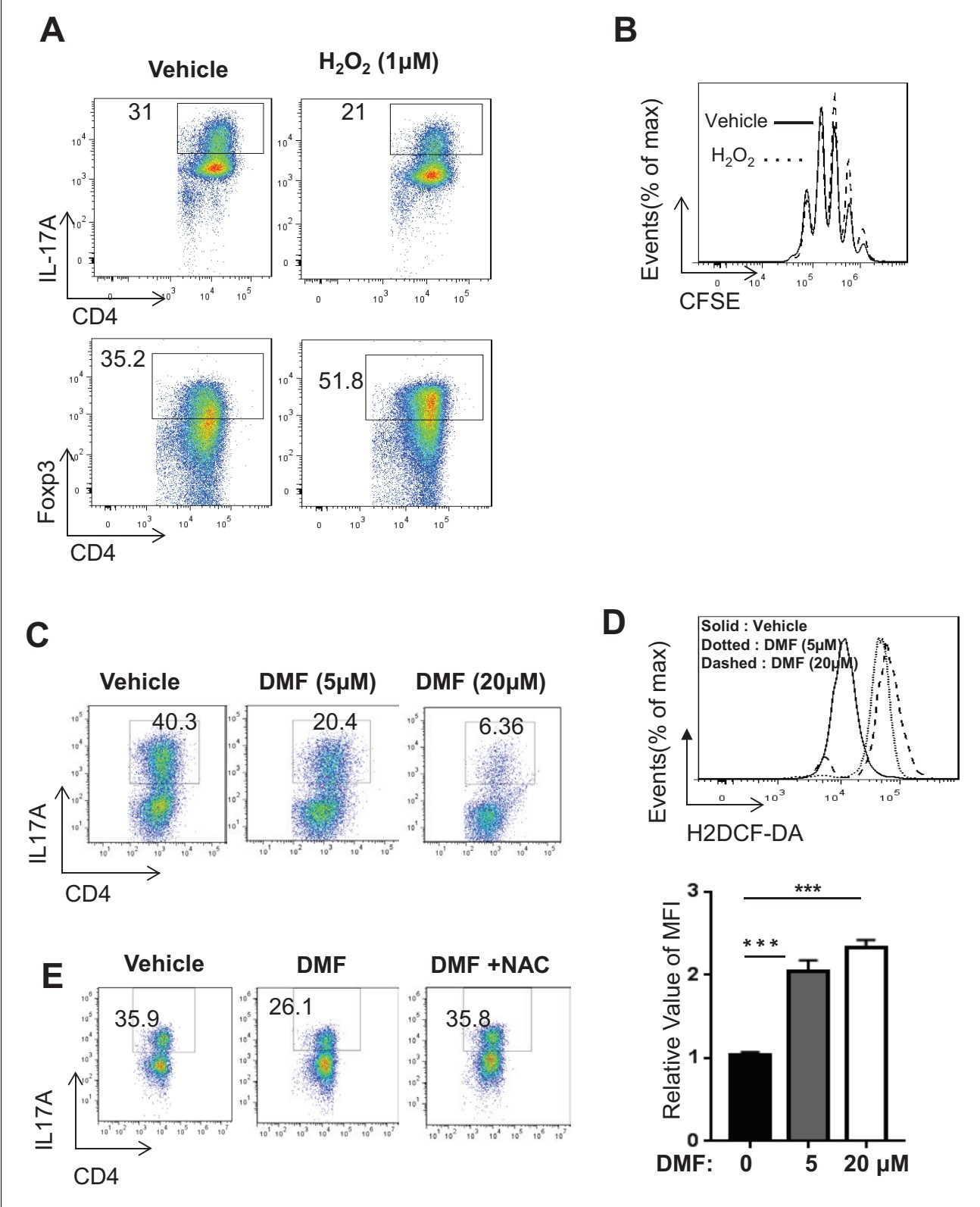

**Figure 5.** DMF suppresses $T_H17$ differentiation by augmenting ROS generation. (**A**) Naive $CD4^+$ T cells from C57BL/6 mice were differentiated under $T_H17$ or $iT_{reg}$-inducing conditions with or without $H_2O_2$ (1 µM) for 5 days, followed by intracellular staining of IL-17 and Foxp3. (**B**) Cell proliferation of active $CD4^+$ T cells (72 hr) with or without $H_2O_2$ (1 µM) was determined as CFSE dilution. (**C–D**) Naive $CD4^+$ T cells from C57BL/6 mice were differentiated under $T_H17$-inducing conditions with indicated dose of DMF for 5 days, followed by intracellular staining of IL-17 (**C**) and ROS (**D**). (**E**)

*Figure 5 continued on next page*

Figure 5 continued

Naive CD4$^+$ T cells from C57BL/6 mice were differentiated under T$_H$17-inducing conditions with indicated treatment for 5 days, followed by intracellular staining of IL-17. Data in **Figure 5** are representative of two-three independent experiments. Data represent the mean ±S.D.
DOI: https://doi.org/10.7554/eLife.36158.017
The following source data and figure supplement are available for figure 5:

**Source data 1.** Source data for D.
DOI: https://doi.org/10.7554/eLife.36158.019
**Figure supplement 1.** DMF suppresses T$_H$17 differentiation through augmenting ROS generation.
DOI: https://doi.org/10.7554/eLife.36158.018

accord with this data, glutamine starvation reduced GSH content in a time-dependent manner in T$_H$17 cells (**Figure 6—figure supplement 1A**). In addition to support GSH biosynthesis, glutamine-derived glutamate can also feed into the TCA cycle. We therefore utilized radiochemical-based approaches to assess activities of glutamine oxidation through the TCA cycle in T$_H$17 and iT$_{reg}$ cells. T$_H$17 cells displayed higher glutamine oxidation activity, indicated by $^{14}CO_2$ release from [U-$^{14}$C]-glutamine, than did iT$_{reg}$ cells (**Figure 6C**). However, mitochondria-dependent pyruvate oxidation through the TCA cycle, indicated by $^{14}CO_2$ release from [2-$^{14}$C]-pyruvate, was comparable between the two functional subsets (**Figure 6C**). These data further suggested that the overall uptake and consumption of glutamine, including oxidation of glutamate through the TCA cycle and utilization of glutamate for GSH biosynthesis is enhanced in T$_H$17 comparing to iT$_{reg}$ cells. Also, qPCR analyses revealed marked upregulation of genes encoding various molecules involved in glutamine catabolism and GSH metabolism in T$_H$17 compared to iT$_{reg}$ cells (**Figure 6—figure supplement 1B and C**). Consistent with this, deprivation of glutamine significantly suppressed T$_H$17 but moderately enhanced iT$_{reg}$ cell differentiation (**Figure 6D**), while both T$_H$17 and iT$_{reg}$ differentiation required glucose (**Figure 6E**). Next, we asked if pharmacological inhibition of the rate-limiting glutaminolyic enzyme glutaminase (Gls) could impact T cell differentiation. Two Gls1 specific inhibitors, bis-2-(5-phenylacetamido-1,2,4-thiadiazol-2-yl) ethyl sulfide (BPTES) and CB-839 (**Wang et al., 2010**; **Le et al., 2012**), slightly enhanced IL-17 expression (**Figure 6—figure supplement 1D**). However, 6-diazo-5-oxo-l-norleucine (DON), an analog of glutamine with broad inhibitory effects glutamine utilizing enzymes (**Pinkus, 1977**; **Shapiro et al., 1979**), skewed away T cells from T$_H$17 toward iT$_{reg}$ differentiation (**Figure 6—figure supplement 1D**). In addition, DON but not BPTES and CB-839 significantly enhanced ROS production under T$_H$17-polarizing condition (**Figure 6—figure supplement 1E**). These results suggested that other glutamine utilizing enzymes including glutamine-dependent amidotransferase and Gls2, the latter of which has been shown to be induced upon T cell activation (**Wang et al., 2011**; **Zalkin and Smith, 1998**; **Massière and Badet-Denisot, 1998**). Glutamine catabolism is not only coupled to the de novo synthesis of GSH, but also generates the anaplerotic substrate, α-ketoglutarate (α-KG), and substrates for nucleotide biosynthesis (**Altman et al., 2016**). A previous study demonstrated that glutamine deprivation enhances iTreg differentiation and addition of α-KG could compromise such effect (**Klysz et al., 2015**). In line with this report, addition of either hypoxanthine and thymidine (HT) or α-KG was able to partially rescue T$_H$17 differentiation in glutamine-free condition without impacting ROS level (**Figure 6F and G**). While NAC treatment is sufficient to suppress ROS production, it was incapable of rescuing T$_H$17 differentiation (**Figure 6—figure supplement 1G**). However, the combination of HT and NAC led to more differentiated T$_H$17 cells and lower level of ROS comparing to HT treatment alone in the absence of glutamine (**Figure 6—figure supplement 1F**). Taken together, our studies suggest that glutamine catabolism directs the lineage choices between T$_H$17 and iT$_{reg}$ cells through supporting T cell proliferation by providing biosynthetic precursors. In addition, glutamine-derived glutamate provides a key substrate for the de novo synthesis of GSH, modulates ROS signaling, and may also impact T cell differentiation (**Figure 6—figure supplement 2**).

## Discussion

A robust T cell-mediated adaptive immune response results from the clonal expansion of antigen-specific T cells and subsequent differentiation into diverse functional subsets to fine-tune responses against challenge. Both the cellular proliferation during expansion and the cytokine production

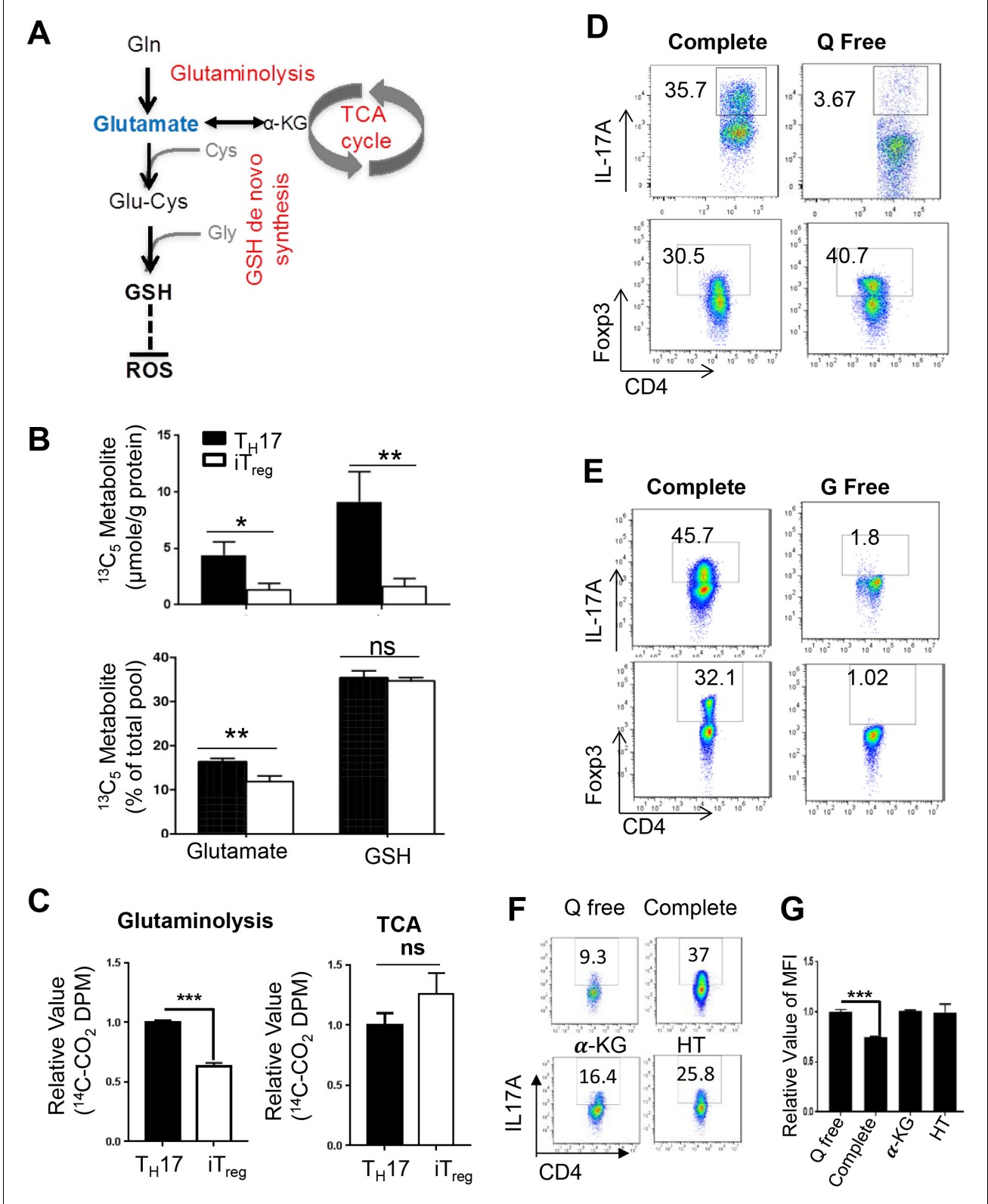

**Figure 6.** Glutamate that fuels GSH de novo synthesis is partially derived from glutamine in T cells. (**A**) Diagram of metabolic steps linked to the GSH production, with metabolic pathways highlighted in red. (**B**) Naive CD4+ T cells from C57BL/6 mice were differentiated under T$_H$17 and iT$_{reg}$–inducing conditions for 5 days, followed by culturing in media containing $^{13}C_5^{15}N_2$-glutamine. The intracellular levels of Glutamate and GSH including $^{13}C$-, $^{13}C$,$^{15}N$-, and $^{12}C$-unlabeled forms were determined by IC-UHRFTMS. (**C**) Naive CD4+ T cells from C57BL/6 mice were differentiated under T$_H$17 or iT$_{reg}$

*Figure 6 continued on next page*

*Figure 6 continued*

cell–inducing conditions for 5 days, were used for measuring the generation of $^{14}CO2$ from [U-$^{14}$C]-glutamine (glutaminolysis), from [2-$^{14}$C]-pyruvate (TCA). (D–E) Naive CD4$^+$ T cells from C57BL/6 mice were differentiated in completed, glutamine-free (Q free) or glucose-free (G free) medium under T$_H$17 or iT$_{reg}$ cell-inducing conditions for 5 days, followed by intracellular staining of IL-17 and Foxp3. (F) Naive CD4$^+$ T cells from C57BL/6 mice were activated in complete medium for 24 hr and cells were washed with PBS and switch to conditional medium in presence or absence of glutamine, 3 mM αKG or 100 μM hypoxanthine and 16 μM thymidine (HT) for 5 days followed by intracellular staining of IL-17 and (G) and ROS. Data represent the mean ±S.D.

DOI: https://doi.org/10.7554/eLife.36158.020

The following source data and figure supplements are available for figure 6:

**Source data 1.** Source data for B, C and G.
DOI: https://doi.org/10.7554/eLife.36158.024
**Figure supplement 1.** Glutamine catabolism is required for driving T$_H$17 and iT$_{reg}$ cell differentiation.
DOI: https://doi.org/10.7554/eLife.36158.021
**Figure supplement 1—source data 1.** Source data for A, B, C, E and G.
DOI: https://doi.org/10.7554/eLife.36158.022
**Figure supplement 2.** Glutamine catabolism coordinates with GSH metabolism in modulating ROS homeostasis and T cell differentiation.
DOI: https://doi.org/10.7554/eLife.36158.023

associated with differentiation exert high bioenergetic and biosynthetic demand on T cells. Accordingly, rapidly evolving pathogens often impose selective pressures on the regulation of central metabolic pathways that fuel cell proliferation and differentiation, allowing T cells to maintain homeostasis while remaining ready to mount rapid responses under diverse metabolic and immune conditions. The inability to accommodate the metabolic and bioenergetic demands of T cell proliferation and differentiation can impair the proper development and function of T cells. Beyond this, the availability of specific metabolites, and the pathways that process them, interconnect with signaling events in the cell to orchestrate metabolic checkpoints which influence T cell activation, differentiation, and immune function (*Wang and Green, 2012*; *Bensinger and Tontonoz, 2008*; *Gerriets and Rathmell, 2012*; *Pearce et al., 2013*; *Chi, 2012*; *Powell and Delgoffe, 2010*; *Michalek and Rathmell, 2010*; *Ho et al., 2015*). Here, we found that glucose plays an indispensable role in driving T$_H$17 and iT$_{reg}$ cell differentiation, while glutamine is only required for T$_H$17 cell differentiation. Glutamine catabolism is coupled with de novo GSH synthesis and is preferentially elevated during T$_H$17 but not iT$_{reg}$ cell differentiation, suggesting that different T cell lineages 'wire' metabolism differently to support their development and function. Previous studies have suggested that engagement of the transcription factor hypoxia-inducible factor 1α (HIF1α) coordinates glycolysis and T cell signaling cascades to regulate the differentiation of T$_H$17 and iTreg cells (*Shi et al., 2011*; *Dang et al., 2011*). While the high rate of glutamine catabolism ensures the capacity to supply glutamate, glycolysis provides ATP and biosynthetic precursors for glycine, fulfilling the needs of de novo synthesis of GSH during T cell differentiation. Our results using genetic modulation of metabolic enzymes suggest that de novo synthesis of GSH but not recycling from GSSG is required for fine-tuning ROS and directing the differentiation of T$_H$17 and T$_{reg}$ cells. We further found that dimethyl fumarate, a FDA approved drug (BG-12/Tecfidera) for multiple sclerosis, suppresses T$_H$17 differentiation by augmenting intracellular ROS. While previous studies clearly demonstrate that activation-induced metabolic reprogramming is required for driving T cell growth and proliferation, our studies shed light on the complex utilization of the glutamine catabolic pathway and implicate ROS as essential metabolic signals that dictate T cell lineage engagement.

Prokaryotic and eukaryotic cells have evolved to maintain reducing intracellular conditions by generating reducing equivalents, NADPH and GSH, enabling cells to fine-tune the ROS levels that are required for fulfilling vital cellular functions. Accordingly, oxidative stress occurs if the balance between ROS production and antioxidant capacity is disturbed, thereby leading to the accumulation of oxidized biomolecules. T cell receptor (TCR) engagement induces a rapid generation of mitochondrial ROS from OXPHOS and cytoplasmic ROS from NADPH oxidases (NOXs), a family of plasma membrane associated oxidases (*Sena et al., 2013*; *Kamiński et al., 2012*; *Jackson et al., 2004*). T cells with reduced production of mitochondrial ROS display impaired production of interleukin 2 (IL-2) and antigen-specific proliferation, indicating an essential signaling role for mitochondrial ROS in driving optimal TCR signaling (*Sena et al., 2013*; *Kamiński et al., 2012*). Beyond that, NOX-

dependent ROS also plays a role in modulating TCR signaling and T cell differentiation. T cells with low levels of ROS due to NOX deficiency are skewed toward a $T_H17$ phenotype (*Jackson et al., 2004*; *Purushothaman and Sarin, 2009*; *Padgett and Tse, 2016*; *Tse et al., 2010*). Our studies suggest that de novo synthesis of GSH is essential for fine-tuning ROS levels in T cells and also indicate the differential requirements for the level of ROS in driving T cell lineage engagement toward $T_H17$ or $iT_{reg}$ cells. Understanding the metabolic process of GSH synthesis and ROS generation during T cell differentiation may also impact the development of safer and more effective therapies for autoimmune and inflammation diseases. Dimethyl fumarate (DMF) is a cellular permeable analog of fumarate and the key active ingredient of BG-12/TECFIDERA and FUMADERM, a first-line oral treatment for relapsing multiple sclerosis (*CONFIRM Study Investigators et al., 2012*; *ALLEGRO Study Group et al., 2012*). However, the cellular and molecular mechanisms of DMF therapy remain largely elusive. DMF has been suggested to impact a plethora of cellular and molecular targets, including Nrf2 and GPCRs in T cells, dendritic cells (DCs), endothelial cells, and neurons (*Blewett et al., 2016*; *Schulze-Topphoff et al., 2016*; *Wang et al., 2015*; *Chen et al., 2014*; *Peng et al., 2012*; *Linker and Gold, 2013*; *Duffy et al., 1998*; *Gill and Kolson, 2013*). Consistent with the studies showing that DMF can neutralize GSH in vitro (*Zheng et al., 2015*; *Sullivan et al., 2013*), our data indicate that the ability of DMF to augment ROS and suppress $T_H17$ cell differentiation contributes to its therapeutic effect.

We found that DMF treatment induced a compensatory NRF2-mediated anti-oxidative response and NRF2 deficiency exacerbated the ROS-producing effects of DMF in $T_H17$ cells. Also, NRF2 and its target genes were highly upregulated in $T_H17$ cells, suggesting that $T_H17$ cells preferentially maintain a low degree of oxidative stress partially by engaging the NRF2 signaling pathway. In addition to GSH production and consumption, NRF2 controls thioredoxin (TXN) production and consumption through transcriptional regulation of its target genes including thioredoxin reductase 1 (TXNRD1). Both GSH and TXN are important anti-oxidation systems and appear functionally redundant in most organisms (*Couto et al., 2016*; *Mustacich and Powis, 2000*; *Lei et al., 2016*; *Lu and Holmgren, 2014*). However, recent studies implicate the presence of a dynamic cross-talk between these two systems. Glutaredoxin (GRX)-GSH can reduce oxidized TXN in the absence of thioredoxin reductase. Conversely, TXN-TXNRD can also function as an alternative system to reduce GSSG to GSH (*Du et al., 2012*; *Tan et al., 2010*; *Johansson et al., 2004*; *Hanschmann et al., 2010*). Collectively, these data indicate that the TXN and GSH systems can backup for each other under certain conditions. Consistent with the overlapping function of the GSH and TXN system, the inhibition of TXNRD rendered cancer cells susceptible to the depletion of GSH (*Mandal et al., 2010*; *Lu et al., 2007*; *Wang et al., 2012*). Although our data suggest that the de novo synthesis of GSH is sufficient to maintain ROS homeostasis in the absence of Gsr dependent GSSG-GSH recycling, it is still conceivable that thioredoxin reductase may partially compensate for the loss of Gsr by backing up the recycle of GSSG to GSH in T cells. It has been shown that TXN can be secreted by $CD4^+$ T cells and may modulate the expression of T cell surface receptor and proliferation (*Rosen et al., 1995*; *Wakasugi et al., 1990*; *Matthias et al., 2002*; *Tagaya et al., 1990*). As such, we envision that the combination of DMF and pharmacological approaches that target TXN system may represent a more effective strategy than DMF alone for treating T cell-mediated inflammation and autoimmune disease.

## Materials and methods

**Key resources table**

| Reagent type (species) or resource | Designation | Source or reference | Identifiers | Additional information |
|---|---|---|---|---|
| Strain, strain background (Mus musculus) | C57BL/6 (B6) mice | | Taconic | |
| Strain, strain background (Mus musculus) | C3H/HeN | | Envigo | |

*Continued on next page*

*Continued*

| Reagent type (species) or resource | Designation | Source or reference | Identifiers | Additional information |
|---|---|---|---|---|
| Genetic reagent (Mus musculus) | *CD4-Cre* *Gclc^{flox/flox}* | PMID:23226398 | | |
| Genetic reagent (Mus musculus) | *Gclm*-KO | PMID:12384496 | | |
| Genetic reagent (Mus musculus) | *ROSA26CreERT2* | RRID: IMSR_JAX:008463 | The Jackson Laboratory | |
| Genetic reagent (Mus musculus) | *GSR*-KO | PMID: 10218442 | | |
| Antibody | Mouse anti-CD3 mAb | Cat. #:BE0001-1, RRID:AB_1107634 | BioXcell | |
| Antibody | Mouse anti-CD28 mAb | Cat. #BE0015-1 RRID:AB_1107624 | BioXcell | |
| Antibody | Mouse anti-IL2 mAb | Cat.#BE0043 RRID::AB_1107702 | BioXcell | |
| Antibody | Mouse anti-IL4 mAb | Cat. #BE0045 RRID:AB_1107707 | BioXcell | |
| Antibody | Mouse anti-IFNγ mAb | Cat. #BE0055 RRID:AB_1107694 | BioXcell | |
| Antibody | Anti mouse CD4-FITC | Cat. #11–0042 RRID:AB_464897 | eBioscience | (1:200) |
| Antibody | Anti mouse CD4-APC | Cat. #17-0041-81 RRID:AB_469319 | eBioscience | (1:200) |
| Antibody | Anti mouse CD8-APC-Cy7 | Cat. #100714 RRID:AB_312753 | Biolegend | (1:200) |
| Antibody | Anti mouse Foxp3-APC | Cat. # RRID:AB_469456 | eBioscience | (1:200) |
| Antibody | Anti mouse IL-17A-PECy7 | Cat. #25-7177-82 RRID:AB_10732356 | eBioscience | (1:200) |
| Antibody | Anti GCLC antibody (rabbit monoclonal) | Cat. #ab190685 RRID:AB_10975474 | Abcam | WB (1:1000) |
| Antibody | Anti GCLM antibody (rabbit monoclonal) | Cat. #ab124827 RRID:AB_10975474 | Abcam | WB (1:1000) |
| Antibody | anti-mouse CD25 -PE | Cat. #101904 RRID:AB_312847 | Biolegend | (1:200) |
| Antibody | anti-mouse CD69-PECy7 | Cat. #552879 RRID:AB_394508 | BD Bioscience | (1:200) |
| Antibody | Anti mouse monoclonal CD3 | Cat. #sc-101442 RRID:AB_1120355 | Santa Cruz | IHC (1:50) |
| Antibody | Anti mouse monoclonal galectin-3 | Cat. #sc-32790, RRID:AB_627657 | Santa Cruz | IHC (1:50) |
| Peptide, recombinant protein | MOG35-55 peptide | synthesized and HPLC-purified | St. Jude Hartwell Center for Biotechnology | |
| Peptide, recombinant protein | Recombinant mouse IL-6 | 216–16 | Peprotech | |
| Peptide, recombinant protein | Recombinant human TGFb | 100–21 c | Peprotech | |

*Continued on next page*

*Continued*

| Reagent type (species) or resource | Designation | Source or reference | Identifiers | Additional information |
|---|---|---|---|---|
| Peptide, recombinant protein | Recombinant human or mouse IL-2 | 200–02 | Peprotech | |
| Commercial assay or kit | Foxp3/Transcription Factor Staining Buffer Set | 00-5523-00 | e-Bioscience | |
| Commercial assay or kit | Naive CD4 + T cell isolation kit,mouse | 5160725186 | Miltenyi Biotec | |
| Commercial assay or kit | CD45R(B220) microbeads, mouse | 5150309030 | Miltenyi Biotec | |
| Commercial assay or kit | ABC kit | PK-7200 | Vector laboratories | |
| Commercial assay or kit | MojoSort Mouse naive CD4 T Cell Isolation Kit | 480031 | Biolegend | |
| Chemical compound, drug | Diethly Fumerate | Sigma Aldrich | D95654 | |
| Chemical compound, drug | N-Acetyl-L-cysteine | Sigma-Aldrich | A7250 | |
| Chemical compound, drug | Tamofixen | Sigma-Aldrich | T5648 | |
| Chemical compound, drug | 4-hydroxytamoxifen | Sigma | H7904 | |
| Chemical compound, drug | Dimethy a-keto glutarate/aKG | Sigma-Aldrich | 34963–1 | |
| Chemical compound, drug | Hypoxathine | Sigma-Aldrich | H9377 | |
| Chemical compound, drug | Thymidine | Sigma | T9250 | |
| Chemical compound, drug | H2O2 | Sigma-Aldrich | 7722-84-1 | |
| Chemical compound, drug | carboxyfluorescein diacetate succinimidyl ester(CFSE) | Invitrogen | C1157 | |
| Chemical compound, drug | DM-H2DCFDA | Invitrogen | C6827 | |
| Chemical compound, drug | DAB | Vector Laboratories | SK-4100 | |
| Chemical compound, drug | Monobromobimane | Invitrogen | M1378 | |
| Chemical compound, drug | 7-amino-actinomycin D(7AAD) | Biolegend | 420404 | |
| Chemical compound, drug | Pertussis toxin | 181 | List Biological Laboratories | |
| Chemical compound, drug | Mycobacterium tuberculosum | 231141 | Difco | |
| Chemical compound, drug | Incomplete Freund's Adjuvant | 263910 | Difco | |
| Chemical compound, drug | [U-14C]-glutamine | MC 1124 | Moravek | |
| Chemical compound, drug | [2–14C]-pyruvate | ARC 0222 | American Radiolabeled Chemicals | |

*Continued on next page*

*Continued*

| Reagent type (species) or resource | Designation | Source or reference | Identifiers | Additional information |
|---|---|---|---|---|
| Chemical compound, drug | Cell Stimulation Cocktail (plus protein transport inhibitors) (500X) | 00-4975-93 | eBioscience | |
| Chemical compound, drug | Iscove's Modified Dulbecco's Media - Glucose free conditional medium | ME17058P1 | Thermo Fisher Scientific | |
| Chemical compound, drug | Iscove's Modified Dulbecco's Media - without L-glutamine | 12–726 f | Lonza | |
| Chemical compound, drug | RPMI 1640 Medium, No Glucose | 11-879-020 | Gibco | |
| Chemical compound, drug | Hyclone RPMI 1640 Medium, no glutamine | sh30096.10 | Thermo Fisher Scientific | |
| Chemical compound, drug | U-13C6-glutamine | CNLM-1275–0.1 | Cambridge Isotope Lab | |
| Chemical compound, drug | 6-Diazo-5-oxo-L-norleucine | D2141-5MG | Sigma-aldrich | |
| Chemical compound, drug | Bis-2-(5-phenylacetamido-1,3,4-thiadiazol-2-yl) ethyl sulfide (BPTES) | SML0601 | Sigma-aldrich | |
| Chemical compound, drug | CB-839 | 22038 | Cayman | |
| Software, algorithm | Graphpad Prism | | RRID:SCR_002798 | |
| Software, algorithm | FlowJo | | RRID:SCR_008520 | |

## Mice

 *Gsr*-KO mice are on C3H/HeN background and *ROSA26CreERT2, CD4-Cre, Gclm*-KO, and *Gclc$^{flox/flox}$* are on the C57BL/6 background and were previously described (*Wang et al., 2011*; *Chen et al., 2007*; *Yang et al., 2002*; *Rogers et al., 2004*; *Pretsch, 1999*; *Yan et al., 2012*) C57BL/6 mice were purchased from Envigo (formly Harlan). Mice at 8–12 weeks of age were used in the experiment and were kept in specific pathogen-free conditions within the Animal Resource Center at the Research Institute at Nationwide Children's Hospital or St. Jude Children's, Research Hospital. Animal protocols were approved by the Institutional Animal Care and Use Committee of the Research Institute at Nationwide Children's Hospital or St. Jude Children's Research Hospital.

## Flow cytometry

For analysis of surface markers, cells were stained in PBS containing 2% (wt/vol) BSA and the appropriate antibodies from eBioscience. Foxp3 staining was performed according to the manufacturer's instructions (eBioscience). For IL-17A intracellular cytokine staining, T cells were stimulated for 4-5 h with phorbol 12-myristate 13-acetate (PMA) and ionomycin in the presence of monensin before being stained according to the manufacturer's instructions (BD Bioscience). For ROS measurement, cells were cultured in fresh serum-free IMDM media containing 5 µM H$_2$DCF-DA (BD Bioscience) for 30 min at 37°C before being washed and resuspended with serum-free IMDM media. The fluorescence intensity was measured by flow cytometry. For GSH measurement, cells were cultured in PBS (1%FBS) containing 50µM monobromobimane (Biochemika) for 10 min at 37°C before being washed and resuspended with PBS. The fluorescence at 450/50 nm (blue spectra) was measured by flow cytometry (*Cossarizza et al., 2009*). Flow cytometry data were acquired on Novocyte (ACEA Biosciences) or LSRII (Becton Dickinson) and were analyzed with FlowJo software (TreeStar).

## Cell purification and culture

Total T cells or naive CD4[+] T cells were enriched from spleens and lymph nodes by negative selection using MACS systems (Miltenyi Biotec, Auburn, CA) following the manufacturer's instructions. Freshly isolated total T cells with 75-80% CD3 positivity were either maintained in culture media with 5ng/ml IL7 or were stimulated with IL-2 (100U/ml) and plate-bound anti-CD3 (clone 145-2C11) and anti-CD28 (clone 37.51). Plates were pre-coated with 2 µg/ml antibodies overnight at 4°C. Cells were cultured in RPMI 1640 media supplemented with 10% (v/v) heat-inactivated fetal bovine serum (FBS), 2 mM L-glutamine, 0.05 mM *2-mercaptoethanol*, 100 units/ml penicillin and 100 µg/ml streptomycin at 37 °C in 5% CO2. For CFSE dilution analysis, cells were pre-incubated for 10 min in 4 µM CFSE (Invitrogen) in PBS plus 5% FBS before culture. For iT$_{reg}$ cell differentiation, 0.5 x10$^6$ naive CD4[+] T cells were stained with 4 µM CFSE and cultured with 100 U/ml IL-2, 5ng/ml TGF-β and 20ng/ml IL-6 in 0.5ml RPMI-1640 media (containing 10% (v/v) heat-inactivated fetal bovine serum (FBS), 2 mM L-glutamine, 0.05 mM *2-mercaptoethanol*, 100 units/ml penicillin and 100 µg/ml streptomycin) in 48-well tissue culture plate that was pre-coated with 10µg/ml anti-CD3 and 10µg/ml anti-CD28 overnight at 4°C . For T$_H$17 conditions, 0.5 x10$^6$ naive CD4[+] T cells and 5 x10$^6$ irradiated splenocytes (artificial APC) were cultured with 2 µg/ml anti-CD3 (2C11; Bio X Cell), 2 µg/ml anti-CD28 (37.51; Bio X Cell), 8 µg/ml anti–IL-2, 8 µg/ml anti–IL-4, 8 µg/ml anti–IFN-γ, 2 ng/ml TGF-β, and 20-50ng/ml IL-6 in 1ml IMDM media (containing 15% (v/v) heat-inactivated fetal bovine serum (FBS), 2 mM L-glutamine, 0.05 mM *2-mercaptoethanol*, 100 units/ml penicillin and 100 µg/ml streptomycin) in 24-well tissue culture plate). For metabolic starvation experiment, Glucose or Glutamine-free IMDM and RPMI-1640 medium was supplemented with 10% (v/v) heat-inactivated dialyzed fetal bovine serum (DFBS). For T$_H$17 rescue experiments, 0.5 x10$^6$ naive CD4+ T cells were activated in complete IMDM medium. After 24 h cells were switched to conditional glutamine-free IMDM medium for 4 days. DFBS was made dialyzing against 100 volumes of distilled water (six changes in three days) using Slide-ALyzer G2 dialysis cassettes with cut-through MW size 2K (ThermoFisher Scientific) at 4$^0$C.

## qPCR and immunoblot analysis.

Total RNA was isolated using the RNeasy Mini Kit (Qiagen) and was reverse transcribed using random hexamer and M-MLV Reverse Transcriptase (Invitrogen). SYBR green-based quantitative RT-PCR was performed using the Applied Biosystems 7900 Real Time PCR System. The relative gene expression was determined by the comparative $C_T$ method also referred to as the $2^{-\Delta\Delta C}{}_T$ method. The data were presented as the fold change in gene expression normalized to an internal reference gene (beta2-microglobulin) and relative to the control (the first sample in the group). Fold change=$2^{-\Delta\Delta C}{}_T$=[($C_{Tgene\ of\ interst}$- $C_{Tinternal\ reference}$)]sample A-=[($C_{Tgene\ of\ interst}$- $C_{Tinternal\ reference}$)]sample B. Samples for each experimental condition were run in triplicated PCR reactions. Primer sequences were obtained from Primer Bank (*Spandidos et al., 2010*). Primer sequences are listed in *Supplementary file 1*. Cell extracts were prepared and immunoblotted as previously described (*Wang et al., 2011*).

## MOG immunization and EAE

Mice were immunized with 100 µg of myelin oligodendrocyte glycoprotein (MOG)$_{35–55}$ peptide in CFA (Difco) with 500 µg of Mycobacterium tuberculosis (Difco). Mice were i.p. injected 200 ng of pertussis toxin (List Biological,#181) on the day of immunization and 2 days later, as described (*Kang et al., 2010*). The mice were observed daily for clinical signs and scored as described previously (*Shi et al., 2011*).

## Histopathology and immunohistochemistry

Mice were euthanized and then were perfused with 25 ml PBS with 2 mM EDTA by heart puncture to remove blood from internal organs. Spinal cords were taken out and fixed by immersion with 10% neutral buffered formalin solution and decalcified. Spinal column was divided into cervical, thoracic and lumbar, and then was embedded in paraffin, sectioned, and stained with standard histological methods for hematoxylin and eosin (H and E). Immunohistochemistry were performed on serial histological sections according to standard protocols using anti-Mac2 and anti-CD3 (1:50, Santa Cruz). Appropriate horseradish peroxidase (HRP)-conjugated secondary antibodies were used and

detected using 3,3'-diaminobenzidine tetrahydrochloride (DAB). Slides were counterstained with hematoxylin. Microscopy images were taken using Zeiss Axio Scope A1.

## Metabolic activity analysis

Glutamine oxidation activity was determined by the rate of $^{14}CO_2$ released from [U-$^{14}$C]-glutamine (*Brand et al., 1984*). In brief, one-five million T cells were suspended in 0.5 ml fresh media. To facilitate the collection of $^{14}CO_2$, cells were dispensed into 7 ml glass vials (TS-13028, Thermo) with a PCR tube containing 50 µl 0.2M KOH glued on the sidewall. After adding 0.5 µci [U-$^{14}$C]-glutamine, the vials were capped using a screw cap with rubber septum (TS-12713, Thermo). The assay was stopped 2 hr later by injection of 100 µl 5N HCL and the vials were kept at room temperate overnight to trap the $^{14}CO_2$. The 50 µl KOH in the PCR tube was then transferred to scintillation vials containing 10 ml scintillation solution for counting. A cell-free sample containing 0.5 µci [U-$^{14}$C]-glutamine was included as a background control.

Pyruvate oxidation activity was determined by the rate of $^{14}CO_2$ released from [2-$^{14}$C]-pyruvate (*Willems et al., 1978*). In brief, one to five million T cells were suspended in 0.5 ml fresh T cell media. To facilitate the collection of $^{14}CO_2$, cells were dispensed into 7 ml glass vials (TS-13028, Thermo) with a PCR tube containing 50 µl 0.2M KOH glued on the sidewall. After adding 0.5 µci [2-$^{14}$C]-pyruvate, the vials were capped using a screw cap with rubber septum (TS-12713, Thermo). The assay was stopped 2 hr later by injection of 100 µl 5N HCL and the vials were kept at room temperate overnight to trap the $^{14}CO_2$. The 50 µl KOH in PCR tube was then transferred to scintillation vials containing 10 ml scintillation solution for counting. A cell-free sample containing 0.5 µci [2-$^{14}$C]-pyruvate was included as a background control.

## Metabolite extraction and analysis by ion chromatography-ultra high resolution-Fourier transform mass spectrometry (IC-UHR-FTMS)

Cells were cultured in glutamine-free media with 2 mM $^{13}C_5^{15}N_2$-glutamine (Cambridge Isotope Laboratories) for 24 hr at 37°C and were then washed three times in cold PBS before snap freezing. The frozen cell pellets were homogenized in 60% cold CH3CN in a ball mill (Precellys- 24, Bertin Technologies) for denaturing proteins and optimizing extraction. Polar metabolites were extracted by the solvent partitioning method with a final CH3CN:H2O:CHCl3 (2:1.5:1, v/v) ratio, as described previously (*Fan et al., 2012*). The polar extracts were lyophilized before reconstition in nanopure water and analysis on a Dionex ICS-5000 +ion chromatography interfaced to a Thermo Fusion Orbitrap Tribrid mass spectrometer (Thermo Fisher Scientific) as previously described (*Fan et al., 2016*) using a *m/z* scan range of 80–700. Peak areas were integrated and exported to Excel via the Thermo TraceFinder (version 3.3) software package before natural abundance correction (*Moseley, 2010*). The isotopologue distributions of metabolites were calculated as the mole fractions as previously described (*Lane et al., 2008*). The number of moles of each metabolite was determined by calibrating the natural abundance-corrected signal against that of authentic external standards. The amount was normalized to the amount of extracted protein, and is reported in µmol/g protein. Metabolome quantification of GSH and GSSG were determined by CE-MS that was carried out through a facility service at Human Metabolome Technology Inc., Tsuruoka, Japan.

## Statistical analysis

*P* values were calculated with Student's *t*-test all experiment except EAE experiments, where two way anova test was performed. *P* values smaller than 0.05 were considered significant, with p-values<0.05, p-values<0.01, and p-values<0.001 indicated as *, **, and ***, respectively.

## Acknowledgements

This work was supported by R21AI117547 and 1R01AI114581 from National Institute of Health, V2014-001 from the V-Foundation and 128436-RSG-15-180-01-LIB from the American Cancer Society (RW), K01AA025093 (YC), R24AA022057 (VV), the American Lebanese and Syrian Associated Charities (DG), and Natural Science Foundation of Hunan Province Grant 2018JJ2351(GL), NCI 1P01CA163223-01A1 and NIDDK 1U24DK097215-01A1 (TWMF), 130421-RSG-17-071-01-TBG from the American Cancer Society , R03 CA212802-01A1 (JY) and R21 AI113930 (YL).

# Additional information

## Funding

| Funder | Grant reference number | Author |
|---|---|---|
| Natural Science Foundation of Hunan Province | 2018JJ2351 | Gaojian Lian |
| American Cancer Society | 130421-RSG-17-071-01-TBG | Jun Yang |
| National Institutes of Health | R03 CA212802-01A1 | Jun Yang |
| National Institutes of Health | K01AA025093 | Ying Chen |
| National Institutes of Health | R24AA022057 | Vasilis Vasiliou |
| American Lebanese and Syrian Associated Charities | | Douglas R Green |
| National Institutes of Health | R21 AI113930 | Yusen Liu |
| National Cancer Institute | 1P01CA163223-01A1 | Teresa WM Fan |
| National Institute of Diabetes and Digestive and Kidney Diseases | 1U24DK097215-01A1 | Teresa WM Fan |
| National Institutes of Health | R21AI117547 | Ruoning Wang |
| American Cancer Society | 128436-RSG-15-180-01-LIB | Ruoning Wang |
| National Institutes of Health | 1R01AI114581 | Ruoning Wang |
| V Foundation for Cancer Research | V2014-001 | Ruoning Wang |

The funders had no role in study design, data collection and interpretation, or the decision to submit the work for publication.

## Author contributions

Gaojian Lian, JN Rashida Gnanaprakasam, Data curation, Formal analysis, Methodology, Writing—original draft; Tingting Wang, Ruohan Wu, Xuyong Chen, Lingling Liu, Yuqing Shen, Data curation, Formal analysis; Mao Yang, Jun Yang, Ying Chen, Vasilis Vasiliou, Douglas R Green, Yusen Liu, Resources; Teresa A Cassel, Teresa WM Fan, Data curation, Methodology; Ruoning Wang, Conceptualization, Supervision, Funding acquisition, Investigation, Methodology, Writing—original draft, Writing—review and editing

## Author ORCIDs

JN Rashida Gnanaprakasam http://orcid.org/0000-0002-9086-0949
Ruoning Wang http://orcid.org/0000-0001-9798-8032

## Ethics

Animal experimentation: Animal protocols were approved by the Institutional Animal Care and Use Committee of the Research Institute at Nationwide Children's Hospital (AR13-00055)

## Decision letter and Author response

Decision letter https://doi.org/10.7554/eLife.36158.028
Author response https://doi.org/10.7554/eLife.36158.029

# Additional files

## Supplementary files

• Supplementary file 1. List of primer sequences used for RT-PCR analysis.
DOI: https://doi.org/10.7554/eLife.36158.025

• Transparent reporting form

DOI: https://doi.org/10.7554/eLife.36158.026

## Data availability

All data generated or analysed during this study are included in the manuscript and supporting files.

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
