## [Decision Letter]

Thank you for submitting your article "Glutathione denovo synthesis coordinates with glutaminolysis to control redoxhomeostasis & directs Tcell differentiation" for consideration by *eLife*. Your article has been reviewed by Tadatsugu Taniguchi as the Senior Editor, a Reviewing Editor, and three reviewers. The following individual involved in review of your submission has agreed to reveal his identity: Chi Van Dang (Reviewer #2).

The reviewers have discussed the reviews with one another and the Reviewing Editor has drafted this decision to help you prepare a revised submission.

Summary:

You have investigated the role of glutathione (GSH) de novo biosynthesis during T cell activation and differentiation. Using mouse genetic models, you found that GSH de novo synthesis plays a critical role in T_H_17 differentiation but is less important for T_reg_ polarization. The defect in T_H_17 differentiation is due to the accumulation of ROS. Furthermore, loss of GSH reductase, the enzyme that reduces oxidized GSSG to GSH, does not have any effect on T cell differentiation. Finally, you have connected glutaminolysis, an important metabolic rewiring step in T cell differentiations, to GSH production. You have asked an interesting and important question and found an interesting answer to complement the discovery by Mak et al. (2017). These studies together elucidate the physiological role of glutathione synthesis in T cell differentiation. However, the reviewers felt there were still some conclusions that need to be strengthened for publication in *eLife*.

Essential revisions:

The reviewers would like to prioritize further data on how glutathione de novo biosynthesis/redox contributes to the biological effect of glutaminolysis in T cell differentiation:

1) Can you rescue T_H_17 differentiation or suppress T_reg_ differentiation in the qlutamine free condition by mitigating the ROS through an anti-oxidant, such as NAC? Does pharmacological inhibition of the first step of glutaminolysis at glutaminase (via CB-839 or BPTES, which is documented to increase ROS) could skew away from T_H_17 toward T_reg_ differentiation.

2) Can exogenous supply of other glutamine-derived metabolites (TCA cycle intermediate or nucleotide) rescue T_H_17 differentiation in glutamine-free condition without influencing the redox status?

3) In Figure 6B, you claimed a glutamine-dependent flux into GSH, but M+5 species is minimal compared to the other species, while glutamine depletion causes very profound defect in T_H_17 in Figure 6D. You need to show the levels of GSH during glutamine starvation and should be more careful about conclusions on the significance of glutamine-dependent GSH production on T_H_17 differentiation.

[Editors' note: further revisions were requested prior to acceptance, as described below.]

Thank you for sending your article entitled "Glutathione de novo synthesis but not recycling process coordinates with glutaminolysis to control redox homeostasis and directs murine T cell differentiation" for peer review at *eLife*. Your article is being evaluated by Tadatsugu Taniguchi as the Senior Editor, a Reviewing Editor, and three reviewers.

There was a diversity of opinion among reviewers and we invite you to respond within the next two weeks with an action plan and timetable to respond whether you will undertake below proposed revisions that require would appear to require new experiments within a two month period or do you disagree and prefer to move ahead with textual changes to address the essential and minor comments without new experiments. Based on your response we will then provide a decision letter. If you decide on the latter course it’s very likely the result will be acceptable, but the editors wanted to leave the former option open to you given that you may see it as an opportunity you don't want to miss.

Essential revisions:

The question regarding rescue of glutamine-deprived T_H_17 cells with NAC or with downstream metabolites was only partially addressed. While you showed that hypoxanthine and thymidine could partially rescue T_H_17 without altered ROS, NAC alone could not rescue the phenotype. You missed the opportunity of combining hypoxanthine, thymidine and NAC to test your hypothesis that ROS plays a critical role. Further, your experiments using CB839, BPTES, and DON were not accompanied by any ROS measurements. Specifically, according to the T_H_-Express database, in contrast to your assertion, Gls is expressed across murine T cell subsets. Hence, the lack of effects of specific Gls inhibitors seem surprising. DON is a non-specific glutamine analog that has pleiotropic effects. ROS measurements would help overcome these concerns.

1) Subsection “GCLC deficiency but not GCLM or GSR deficiency suppresses T cell activation and proliferation”, why the authors concluded "T cells are relatively resistant toward ferroptosis", as I saw a modest reduction of viability.

2) Subsection “Glutamate that fuels GSH de novo synthesis is partially derived from glutamine during T_H_17 cell differentiation”, a simplest explanation of Figure 6B is that glutamine uptake in iT_reg_ is significantly lower. But whatever glutamine that cells can uptake, contributes similarly to glutamate and GSH between T_H_17 and iT_reg_.

3) Subsection “Glutamate that fuels GSH de novo synthesis is partially derived from glutamine during T_H_17 cell differentiation”, I understand that the assay has been widely used, but the fundamental rationale behind it is unclear. Because the carbon dioxide released from glutamine is from α-ketoglutarate to succinyl-CoA, a step downstream of the glutamate. I think the experiment proves that glutaminolysis is more active in T_H_17 than iT_reg_ but cannot further justify the contribution of glutamine oxidation to GSH production.

---

## [Author Response]

Essential revisions:The reviewers would like to prioritize further data on how glutathione de novo biosynthesis/redox contributes to the biological effect of glutaminolysis in T cell differentiation:1) Can you rescue T_H_17 differentiation or suppress T_reg_ differentiation in the qlutamine free condition by mitigating the ROS through an anti-oxidant, such as NAC?

We are thankful for the reviewer’s thoughtful assessment and suggestion. We have performed the differentiation experiment to evaluate the effect of NAC on rescuing T_H_17 differentiation in glutamine free condition. We found that NAC treatment was incapable of rescuing T_H_17 differentiation (Figure 6—figure supplement 1F). NAC addition also failed to restore cell proliferation in the absence of glutamine as indicated by comparable low cell numbers in both groups (Figure 6—figure supplement 1G). On the other hand, our conclusion is that GSH de novo biosynthesis requires glutamine-derived glutamate as precursor and plays a critical role in regulating ROS homeostasis during T_H_17 cell differentiation. We apologize if we have given the unintended impression that GSH de novo biosynthesis is responsible for all essential functions of glutaminolysis. Other glutamine-derived metabolites, as the reviewer pointed out below, also play significant roles in driving T cell proliferation, maintaining cell viability and regulating T cell differentiation. We ensured that this is stated clearly in the text.

Does pharmacological inhibition of the first step of glutaminolysis at glutaminase (via CB-839 or BPTES, which is documented to increase ROS) could skew away from T_H_17 toward T_reg_ differentiation.

We are thankful for the reviewer’s insightful suggestion. Glutaminase (GLS) (Altman et al. 2016) is a rate-limiting enzyme in glutamine catabolic pathway. Assessing the impact of GLS inhibitors in modulating T cell differentiation will not only provide mechanistic insights and also provide useful information that may have the potential to direct clinical application in the future. As the reviewer suggested, we have examined T_H_17 and iT_reg_ differentiation in the presence of CB-839 or BPTES, two recently developed GLS inhibitors that preferentially inhibit GLS1 (Wang et al. 2010, Le et al. 2012). In addition, we also include 6-diazo-5-oxo-l-norleucine (DON), an analog of glutamine with broad inhibitory effects glutamine utilizing enzymes (Pinkus 1977, Shapiro et al. 1979), since our previous study indicated that GLS2 is significantly induced following T cell activation (Wang et al. 2011). Our results showed that DON treatment significantly suppressed T_H_17 cell differentiation (Figure 6—figure supplement 1D). Instead, BPTES or CB-839 treatment slightly enhanced IL-17A expression (Figure 6—figure supplement 1D). In line with these data, DON but not BPTES or CB-839 treatment significantly enhanced ROS production under T_H_17-polarizing condition (Figure 6—figure supplement 1E).

2) Can exogenous supply of other glutamine-derived metabolites (TCA cycle intermediate or nucleotide) rescue T_H_17 differentiation in glutamine-free condition without influencing the redox status?

We share a similar thought on the relevance of other glutamine-derived metabolites with the reviewers. Glutamine catabolism is not only coupled to the de novo synthesis of GSH, but also generates the anaplerotic substrate, α-ketoglutarate (α-KG), and substrates for nucleotide biosynthesis. As the reviewer pointed out, it is important to assess the contribution of α-KG and nucleotide in T_H_17 cell’s glutamine dependency. Therefore, we have supplied T cells with either α-KG and or hypoxanthine + thymidine (HT) under glutamine free condition and assessed T cell differentiation and ROS. Our data showed that the addition of either HT or α-KG was able to partially rescue T_H_17 differentiation in glutamine-free condition without impacting ROS level (Figure 6F and 6G). These results clearly demonstrated that other glutamine-derived metabolites, including nucleotides and α-KG contribute to T_H_17 cells’ glutamine dependency independent of ROS regulation. Again, we apologize if we have given the impression that GSH de novo biosynthesis is responsible for all essential functions of glutaminolysis. We ensured that this is stated clearly in the text.

3) In Figure 6B, you claimed a glutamine-dependent flux into GSH, but M+5 species is minimal compared to the other species, while glutamine depletion causes very profound defect in T_H_17 in Figure 6D. You need to show the levels of GSH during glutamine starvation and should be more careful about conclusions on the significance of glutamine-dependent GSH production on T_H_17 differentiation.

We apologize for giving the impression of overstating our data and have adjusted the text to properly explain our data. We also agree with the reviewer’s assessment on M+5 species data and appreciate the reviewer’s constructive comment on including GSH data following glutamine starvation. We have now included the data showing that, glutamine starvation reduced GSH content in a time-dependent manner in T_H_17 cells (Figure 6—figure supplement 1A). In addition, we have optimized the experimental conditions in assessing glutamine metabolic flux using ^13^C-tracer. In addition, we have included iT_reg_ cells group as a comparison of T_H_17 cells (Figure 6B). Our data revealed that T_H_17 cells displayed a higher capacity of funneling glutamine-derived carbons to glutamate and GSH than iT_reg_ cells, as indicated by significantly elevated quantity of ^13^C_5_ labelled glutamate and GSH (Figure 6B, upper panel). Moreover, a significant portion of GSH pool comes from ^13^C_5_-GSH, indicative of a direct conversion of ^13^C-glutamine into GSH (Figure 6B, lower panel).

[Editors' note: further revisions were requested prior to acceptance, as described below.]

Essential revisions:The question regarding rescue of glutamine-deprived T_H_17 cells with NAC or with downstream metabolites was only partially addressed. While you showed that hypoxanthine and thymidine could partially rescue T_H_17 without altered ROS, NAC alone could not rescue the phenotype. You missed the opportunity of combining hypoxanthine, thymidine and NAC to test your hypothesis that ROS plays a critical role. Further, your experiments using CB839, BPTES, and DON were not accompanied by any ROS measurements. Specifically, according to the Th-Express database, in contrast to your assertion, Gls is expressed across murine T cell subsets. Hence, the lack of effects of specific Gls inhibitors seem surprising. DON is a non-specific glutamine analog that has pleiotropic effects. ROS measurements would help overcome these concerns.

We thank the reviewers for the additional constructive comments. We took this opportunity to further strengthen our studies by performing additional experiments and properly interpreting our data.

Specifically,

1) We have assessed T_H_17differentiation in glutamine-starvation condition after adding HT and NAC either alone or in combination. While NAC alone is insufficient to support T_H_17differentiation (likely due to its incapability of supporting T cell proliferation in the absence of glutamine), the combination of NAC and HT resulted in more differentiated T_H_17cells and lower level of ROS comparing to HT alone in the absence of glutamine (Figure 6—figure supplement 1F and G). These results support the idea that glutamine catabolism supports T_H_17differentiation through providing precursors for supporting both proliferation and redox regulation.

2) For CB839, BPTES, and DON experiment, we have included original ROS data (Figure 6—figure supplemental 1E). We found that only DON treatment increases ROS production, which is in consistent with our T_H_17differentiation data (only DON suppresses T_H_17differentiation). Now, we include both histogram and bar-graph in our revised figure. We also thank the reviewer for pointing out the expression data of Gls1 extracted from T_H_-Express database. We apologize for giving the impression that Gls1 is not expressed in CD4^+^ T cells. We agree with the reviewer’s assessment on the pleiotropic effects of DON. We now carefully revised our interpretation on the data. In addition to Gls1, we reasoned that Gls2 may compensate for the inhibition of Gls1. Gls2 is also expressed in all T cell subsets with the highest expression level in T_H_17subset based on T_H_-Express database, and the protein level of Gls2 has been shown to be induced following T cell activation (Wang, Dillon et al., 2011). We also speculated that other glutamine utilizing enzymes including amidotransferase could convert glutamine into glutamate and might be involved in supporting GSH biosynthesis in T cells.

1) Subsection “GCLC deficiency but not GCLM or GSR deficiency suppresses T cell activation and proliferation”, why the authors concluded "T cells are relatively resistant toward ferroptosis", as I saw a modest reduction of viability.

We agree with the reviewer’s assessment and have rephrased our interpretation according to the suggestion.

2) Subsection “Glutamate that fuels GSH de novo synthesis is partially derived from glutamine during T_H_17 cell differentiation”, a simplest explanation of Figure 6B is that glutamine uptake in iT_reg_ is significantly lower. But whatever glutamine that cells can uptake, contributes similarly to glutamate and GSH between T_H_17 and iT_reg_.

We thank the reviewer for this insightful comment and have now included the heightened glutamine update as a possible explanation of our data.

3) Subsection “Glutamate that fuels GSH de novo synthesis is partially derived from glutamine during T_H_17 cell differentiation”, I understand that the assay has been widely used, but the fundamental rationale behind is unclear. Because the carbon dioxide released from glutamine is from α-ketoglutarate to succinyl-CoA, a step downstream of the glutamate. I think the experiment proves that glutaminolysis is more active in T_H_17 than iT_reg_ but cannot further justify the contribution of glutamine oxidation to GSH production.

We apologize for giving the impression of overstating our data and have adjusted the text to properly explain our data according to the reviewer’s comment. Specifically, we have now highlighted the possibility that the overall uptake and consumption of glutamine, including oxidation of glutamate through the TCA cycle and utilization of glutamate for GSH biosynthesis is enhanced in T_H_17 comparing to iT_reg_ cells.

We thank the editor and all of the assigned reviewers for the time and effort involved in assessing our work, and we hope that the reviewer agrees that in addressing the concerns we have substantially improved our manuscript.